# Sparse Polyak: an adaptive step size rule for high-dimensional M-estimation

**Tianqi Qiao**
Texas A&M University
College Station, TX, USA
tianqi.qiao@tamu.edu

**Marie Maros**
Texas A&M University
College Station, TX, USA
mmaros@tamu.edu

## Abstract

We propose and study Sparse Polyak, a variant of Polyak's adaptive step size, designed to solve high-dimensional statistical estimation problems where the problem dimension is allowed to grow much faster than the sample size. In such settings, the standard Polyak step size performs poorly, requiring an increasing number of iterations to achieve optimal statistical precision-even when, the problem remains well conditioned and/or the achievable precision itself does not degrade with problem size. We trace this limitation to a mismatch in how smoothness is measured: in high dimensions, it is no longer effective to estimate the Lipschitz smoothness constant. Instead, it is more appropriate to estimate the smoothness restricted to specific directions relevant to the problem (restricted Lipschitz smoothness constant). Sparse Polyak overcomes this issue by modifying the step size to estimate the restricted Lipschitz smoothness constant. We support our approach with both theoretical analysis and numerical experiments, demonstrating its improved performance.

## 1 Introduction

Consider the high-dimensional statistical estimation problem

$$\min_{\mathbb{R}^d \ni \theta : \|\theta\|_0 \leq s} f(\theta) = \frac{1}{n} \sum_{i=1}^{n} \ell(z_i, \theta), \tag{1}$$

with data points $z_i \in \mathbb{R}^d, i = 1 \dots n$. We focus on the regime in which the dimensionality grows much faster than the sample size, i.e. $\frac{d}{n} \to \infty$. To obtain consistent estimates in this setting, it is necessary to assume that the true solution exhibits some low-dimensional structure-such as sparsity. In (1) sparsity is enforced through the $\ell_0$ constraint, which guarantees that $\theta$ will have at most $s$ non-zero elements. This constraint renders the problem in (1) non-convex and, in general, NP-hard, regardless of the objective function $f$ Natarajan (1995). Nevertheless, under certain assumptions on the data, various algorithms have been developed to efficiently find approximate solutions to (1). Notably, under suitable assumptions that hold for a variety of statistical models, the Iterative Hard Thresholding (IHT) algorithm has been shown to efficiently find sufficiently accurate solutions to (1). The IHT algorithm results from applying projected gradient descent to (1) and reads

$$\theta_{t+1} = \text{HT}_s \left( \theta_t - \gamma \nabla f(\theta_t) \right),$$

where $\text{HT}_s$ denotes the hard thresholding operator. $\text{HT}_s$ retains the $s$ largest-magnitude components of its input and sets the remaining to zero. Here, $\gamma > 0$ denotes the step-size which ought to be chosen as $\gamma = 2/(3\bar{L})$ Jain et al. (2014), where $\bar{L}$ denotes the *restricted* Lipschitz smoothness (RSS) constant, and can be interpreted as the Lipschitz smoothness constant of $f$ when restricted to sparse directions.

39th Conference on Neural Information Processing Systems (NeurIPS 2025).

For a variety of statistical models, such as Generalized Linear Models (GLMs), $\bar{L}$ remains constant as long as $\frac{s \log(d)}{n}$ remains constant, even if $\frac{d}{n} \to \infty$. This insight underpins the *rate invariance* of IHT: under suitable conditions, the number of iterations required to achieve (near) optimal statistical precision remains constant even as both $d$ and $n$ grow.

Analogous to the Lipschitz smoothness constant, the RSS constant must be estimated in practice. Thus, the natural question in this context, and the starting point to the work in this paper is: **(i)** Do already existing approaches to adaptively tune $\gamma$ via the estimation of the Lipschitz smoothness constant work in the high-dimensional setting? Our criteria to determine whether a step-size rule works in the high dimension, additional to convergence will be determined by the answers to the following questions: **(ii)** Can they achieve the same or better guarantees than by choosing the optimal fixed step-size? **(iii)** Can they guarantee rate invariance as $\frac{d}{n} \to \infty$?

## 1.1 Related works

Over the past three decades, numerous methods have been proposed to solve the problem in (1), including Matching Pursuit Mallat and Zhang (1993), Orthogonal Matching Pursuit Pati et al. (1993), and CoSaMP Needell and Tropp (2009). Iterative Hard Thresholding (IHT) was first introduced in Blumensath and Davies (2009) with many variants proposed since. While initial convergence guarantees for IHT required the Restricted Isometry Property (RIP)-a condition often too stringent in practice-Jain et al. (2014) extended IHT's convergence guarantees to problems fulfilling the restricted strong convexity (RSC) and RSS; establishing linear convergence to near optimal statistical precision. Observe that in the M-estimation context, convergence to arbitrary precision is unnecessary and convergence to the best achievable statistical precision is preferred. However, for the results in Jain et al. (2014)

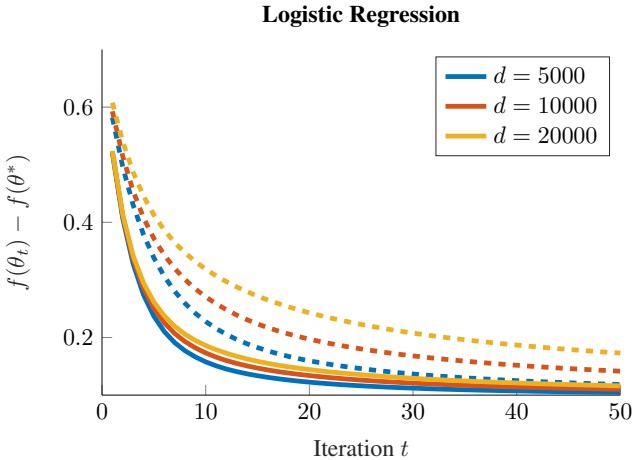

**Logistic Regression**

Figure 1: Performance of Polyak's step size (dashed) and Sparse Polyak (solid) on logistic regression problems with increasing $d$ and $n$. The quantities $s$, $s^*$, $\bar{\kappa}$ and $\frac{\log(d)}{n}$ remain constant. With Polyak's step-size the performance degrades as $d$ increases whereas Sparse Polyak exhibits rate invariance, i.e. the number of iterations to achieve (near) optimal statistical precision does not change.

to hold, if the optimal parameter to recover is $s^*-$sparse, $s \geq \mathcal{O}\left(\bar{\kappa}^2 s^*\right)$ is required, where $\bar{\kappa}$ is the restricted condition number. Khanna and Kyrillidis (2018) proposed an accelerated version of IHT which was extended to the stochastic setting in Zhou et al. (2018). Zhou et al. (2018) establishes that with a sufficiently large mini-batch size, acceleration can be achieved and the faster rate requires only $s \geq \mathcal{O}(\bar{\kappa}s^*)$. Axiotis and Sviridenko (2022) proposed a variant of IHT with an adaptively chosen weighted $\ell_2$ penalty for which only $s \geq \mathcal{O}(\bar{\kappa}s^*)$ is required. They further establish that for IHT $s \geq \mathcal{O}(\bar{\kappa}s^*)$ is in fact a necessary condition to achieve near optimal statistical precision. Li et al. (2016) and Shen and Li (2018) introduced variants of IHT that incorporate variance reduction. Shen and Li (2017b) and Yuan et al. (2018) propose Partial Hard Thresholding and Gradient Hard Thresholding pursuit respectively, with a focus on support recovery under high SNR assumptions. Yuan and Li (2021) establishes generalization bounds for solutions found via IHT. Further, Zhang et al. (2025) establishes that IHT, for a range of $\bar{\kappa}$, finds solutions that can be shown to achieve the oracle estimation rate under a high SNR condition.

With no exception, the discussed works establishing convergence under the RSS require knowledge of the RSS constant $\bar{L}$. Knowledge of $\bar{L}$ is crucial for the requirement $s \geq \mathcal{O}\left(\bar{\kappa}^2 s^*\right)$ being sufficient for convergence to optimal statistical precision. More generally, convergence can be established whenever $s \geq \mathcal{O}\left(\frac{s^*}{\bar{\mu}^2 \gamma^2}\right)$, with $\gamma \leq \frac{1}{\bar{L}}$. Consequently, overestimating $\bar{L}$ forces a smaller step size $\gamma$, which in turn slows down convergence and leads to denser solutions.

Several recent works have proposed adaptive step-size schemes based on estimating the local Lipschitz constant Malitsky and Mishchenko (2020, 2024); Latafat et al. (2024). With the goal to jointly exploit the function's local regularity and the algorithm's trajectory, Mishkin et al. (2024) derive new convergence results for gradient descent as a function of the objective function's local Lipschitz smoothness and strong convexity parameters. They additionally establish that Polyak's step-size obtains fast and path-dependent rates. However, their results do not naturally extend to constrained problems, particularly non-convex ones like (1).

Polyak's original step-size rule, first proposed in Polyak (1969), has gathered renewed attention in the machine learning community Ren et al. (2022); Hazan and Kakade (2019); Loizou et al. (2021); Wang et al. (2023); Zamani and Glineur (2024), but it remains ill-suited for non-convex constrained settings such as (1) unless additional assumptions are imposed. Some efforts have been made to adapt Polyak's step size to constrained problems. For instance, Cheng and Li (2012) addresses box constraints, while Devanathan and Boyd (2024) consider convex constraints.

To this day, except for Li et al. (2024), and works that employ inexact line-search strategies Xiao and Zhang (2013); Wang et al. (2014), there has been no study of adaptive step-size schemes in the high-dimensional context. The work in Li et al. (2024) handles (1) in the stochastic setting and proposes the use of Polyak's step-size with no modification. The results in Li et al. (2024) are limited even with no stochasticity, as they imply bounds on the restricted condition number, i.e. $\bar{\kappa} \leq \frac{\eta}{\eta-1}$ where $\eta = (\sqrt{5}+1)^2/4$. Further, it is worth mentioning that the notion of RSS assumed in Li et al. (2024) is more restrictive than that assumed in the present paper and Jain et al. (2014). Further, as shown in Fig. 1 Polyak's step-size with no modification presents a performance that degrades as the size of the problem increases even if $\bar{\kappa}$ and the optimal statistical precision (of the order of $\mathcal{O}(\frac{s^* \log d}{n})$ for this particular statistical model) remain constant. This effect is highly undesirable in the high-dimensional setting and as shown in the present work, can be avoided with a suitable modification of Polyak's step-size.

## 1.2 Major contributions

Our main contribution is the first adaptive step-size rule that performs well in high-dimensions and preserves the rate invariance property. To develop this scheme, we address question **(i)** posed in the introduction. We observe that adaptive step-size rules that estimate the Lipschitz smoothness constant do not necessarily work well in high-dimensions. This is because, in many common statistical models, the Lipschitz smoothness constant scales as $\mathcal{O}(d)$ with high probability. Consequently, we answer questions **(i)** through **(iii)** in the negative, by demonstrating empirically (c.f. Fig. 1) that estimating the Lipschitz smoothness constant via Polyak's step-size (dashed line) does not yield rate invariance as $d$ grows even if $\frac{\log(d)}{n}$ is held constant.

To overcome this limitation, we design an adaptive step-size rule that estimates the restricted Lipschitz smoothness constant instead. With this modification, we answer questions **(ii)** and **(iii)** in the affirmative. This is captured in Theorem 1 and its Corollaries, which particularize the results of Theorem 1 to relevant statistical models. In Sections 3, 6, and in Appendix D we provide theoretical and empirical evidence, both on synthetic and real data, that our proposed method outperforms Polyak's step-size for high-dimensional M-estimation tasks and achieves rate invariance. Moreover, we theoretically and empirically show that Sparse Polyak converges to optimal statistical precision at least as fast as IHT with the optimal fixed step-size $\gamma = \mathcal{O}(1/\bar{L})$. We also establish sufficient conditions under which we can guarantee support recovery, and particularize our results for specific statistical learning models in Section 4.

Our guarantees are derived under standard assumptions, identical to those in Jain et al. (2014); Shen and Li (2017b); Yuan et al. (2018). To guarantee convergence to optimal statistical precision we require the knowledge of $f(\theta^*)$ where $\theta^*$ denotes the true parameter. While $f(\theta^*)$ is known to be of the order $\mathcal{O}\left(\frac{\log d}{n}\right)$, its exact value is typically unknown. Consequently, we provide in Appendix B a double loop method that estimates a surrogate to $f(\theta^*)$ and allows convergence to optimal statistical precision as long as lower bound to $f(\theta^*)$ is known. Observe that in our context $f(\theta^*) \geq 0$ making 0 a valid lower bound.

In addition, we prove linear convergence for statistical models that do not satisfy the regularity conditions in Jain et al. (2014); Axiotis and Sviridenko (2022); Yuan and Li (2021) but instead fulfill

the weaker condition in Loh and Wainwright (2015). This extends the results in Jain et al. (2014); Yuan and Li (2021) to cover additional GLMs with an adaptive step-size rule. We provide these additional results in Appendix C.2.

Our proof technique may be of independent interest as it provides a clear pathway to establishing convergence of the IHT algorithm. We believe this is key in extending theoretical guarantees to adaptive step-size schemes to solve (1). We provide a sketch of the proof of our main results in Section 5 and the formal proof in Appendix A.

Finally, although our analysis applies only to Polyak's step-size, we conjecture that other adaptive step-size rules such as Barzilai and Borwein (1988), Zhou et al. (2025), Malitsky and Mishchenko (2020) may also suffer from similar performance degradation in high dimensions, and would therefore require analogous re-engineering to be effective.

**Notation.**  Throughout this paper, we adopt the following notations. For vectors $x, \in \mathbb{R}^d$, we denote by $x_i$ the $i^{\text{th}}$ element. We denote the $\ell_\infty$-norm of $x$ as $\|x\|_\infty$, the Euclidean norm as $\|x\|$, the $\ell_1$-norm as $\|x\|_1$, and the $\ell_0$-norm as $\|x\|_0$. Recall $\|x\|_0 = |\{i : x_i \neq 0\}|$. Also, we let $|x|_{\min}$ denote the minimal entry of $x$ in the sense of absolute value. The inner product between two vectors is denoted as $\langle x, y \rangle$. Matrices such as $X \in \mathbb{R}^{d \times d}$, are capitalized. For any matrix $\Sigma$, we denote its largest singular value by $\sigma_{\max}(\Sigma)$ and its smallest singular value by $\sigma_{\min}(\Sigma)$. Similarly, if $\Sigma \in \mathbb{R}^{d \times d}$ diagonalizes, we use $\lambda_{\max}(\Sigma)$ and $\lambda_{\min}(\Sigma)$ to denote the largest and smallest eigenvalues of $\Sigma$ respectively. The Frobenius norm of $X$ is given by $\|X\|_F$, and the nuclear norm by $\|X\|_*$.

## 2   Setup and background

We make the following assumptions regarding the objective function $f$.

**Assumption 1** (RSC Agarwal et al. (2012))**.** *The objective function $f$ is $(\mu, \tau)-$restricted strongly convex in $\mathbb{R}^d$, i.e.*

$$\frac{\mu}{2}\|\theta_1 - \theta_2\|^2 - \frac{\tau}{2}\|\theta_1 - \theta_2\|_1^2 \leq f(\theta_1) - f(\theta_2) - \langle \nabla f(\theta_2), \theta_1 - \theta_2 \rangle, \, \forall \theta_1, \, \theta_2 \in \mathbb{R}^d. \quad (2)$$

**Assumption 2** (RSS Agarwal et al. (2012))**.** *The objective function $f$ is $(L, \tau)-$restricted smooth in $\mathbb{R}^d$, i.e.*

$$f(\theta_1) - f(\theta_2) - \langle \nabla f(\theta_2), \theta_1 - \theta_2 \rangle \leq \frac{L}{2}\|\theta_1 - \theta_2\|^2 + \frac{\tau}{2}\|\theta_1 - \theta_2\|_1^2, \forall \theta_1, \, \theta_2 \in \mathbb{R}^d.$$

Assumptions 1 and 2 extend the classical notions of strong convexity and $L-$Lipschitz smoothness. These assumptions reduce to their classical counterparts when $\tau$ is sufficiently small and the direction $\theta_1 - \theta_2$ is appropriately sparse. Observe that when $\theta_1 - \theta_2$ is dense and $f$ is convex, Assumption 1 becomes vacuous and the upper bound in Assumption 2 scales linearly with the problem dimension $d$. This highlights that the RSC and RSS depend not only on the magnitude of the direction $\theta_1 - \theta_2$ but also its structure.

In high-dimensional statistical learning settings where $\frac{d}{n} \to \infty$, standard strong convexity and smoothness assumptions fail to hold. However , many important problems still satisfy variants of the RSC and RSS, with both $\mu$ and $L$ remaining dimension-independent and with $\tau$ exhibiting only moderate dependence on $d$, e.g., Jain et al. (2014); Agarwal et al. (2012). We leverage this in Section 4 to establish fast computational and near optimal statistical guarantees for a variety of high-dimensional statistical learning problems.

We further highlight that our results can be generalized by adopting a weaker RSC condition, where we assume that (2) holds only for pairs $\theta$, $\theta^*$ satisfying $\|\theta - \theta^*\| \leq 1$, where $\theta$ denotes the ground truth, rather than requiring it to hold globally. This relaxation broadens the applicability of our approach, allowing it to accommodate a wider class of functions. Notably, for some generalized linear models (GLMs), the loss function does not necessarily satisfy (2) without imposing additional constraints. We provide a detailed discussion of these results in Appendix C.2.

## 3   Main Result

In this section we present our main theoretical result, which establishes convergence guarantees for Sparse Polyak (c.f. Algorithm 1). In this section, we focus on the deterministic setting of $f$, while

the statistical case will be addressed in Section 4. Note that IHT corresponds to projected gradient descent when applied to (1). The projection onto the $\ell_0$ is given by the hard thresholding operator, already discussed in the introduction and formally defined as follows.

**Definition 1** (Hard Thresholding Operator). *For any $s > 0$, and $z \in \mathbb{R}^d$, we define $\mathrm{HT}_s(z)$ as the projection of $z$ on $B_0(s) := \{\theta \in \mathbb{R}^d \mid \|\theta\|_0 \leq s\}$. i.e.,*

$$\mathrm{HT}_s(z) = \underset{\theta \in B_0(s)}{\operatorname{argmin}} \|\theta - z\|_2,$$

*where ties are broken lexicographically.*

---

**Algorithm 1** Iterative Hard Thresholding (IHT) with Polyak Step-Size

---

1: **Input:** Function $f$, target function value $\widehat{f}$, sparsity parameter $s$, number of iterations $T$
2: **Initialize:** $\theta_0 \in \mathbb{R}^d$, with $\|\theta_0\|_0 \leq s$
3: **for** t = 0 to $T - 1$ **do**
4:     Compute step-size $\gamma_t = \frac{\max\{f(\theta_t) - \widehat{f}, 0\}}{5\|\mathrm{HT}_s(\nabla f(\theta_t))\|^2}$
5:     Update: $\theta_{t+1} = \mathrm{HT}_s\left(\theta_t - \gamma_t \nabla f(\theta_t)\right)$
6: **end for**
7: **Output:** $\theta_T$

---

The adaptive step-size rule in Algorithm 1 differs from the classical Polyak rule by replacing $\|\nabla f(\theta_t)\|^2$ with $\|\mathrm{HT}_s(\nabla f(\theta_t))\|^2$. This approach contrasts with the low-dimensional case and with the work in Li et al. (2024). Observe that even if the current iterate $\theta_t$ is sparse, sparsity of $\nabla f(\theta_t)$ can not be guaranteed. In fact, the worst-case relationship $\|\nabla f(\theta_t)\| \leq \sqrt{\frac{d}{s}}\|\mathrm{HT}_s(\nabla f(\theta_t))\|$, which holds with equality for a vector in which all coordinates are identical, may hold. As a result, using the full gradient norm can lead to using overly conservative step-sizes, slowing down convergence dramatically as $d$ increases. Unless strong additional conditions are imposed on the problem, convergence may be even be jeopardized (see discussion in Section 5 for more details).

Before providing our main result we introduce some notation. Consider fixed values $s \geq s^* > 0$, and let $f^* \triangleq \min_{\theta : \|\theta\|_0 \leq s^*} f(\theta)$. Assume the chosen target function value satisfies $\widehat{f} \geq f^*$. Let $\bar{L} = L + 3\tau s$, $\bar{\mu} = \mu - 3\tau s$, and $\bar{\kappa} = \bar{L}/\bar{\mu}$ denote the restricted condition number.

**Theorem 1.** *Let $\{\theta_t\}_{t \geq 1}$ denote the sequence of iterates generated by Algorithm 1. Suppose the objective function $f$ satisfies the RSC and RSS in Assumptions 1 and 2, respectively. Let $\widehat{\theta}$ be any $s^*$-sparse vector such that $f(\widehat{\theta}) = \widehat{f}$, and assume $\bar{\mu} > 0$ and $s \geq (240\bar{\kappa})^2 s^*$. Then, for any iterate $\theta_t$ such that $\|\theta_t - \widehat{\theta}\|^2 \geq \frac{36\|\mathrm{HT}_s(\nabla f(\widehat{\theta}))\|^2}{\bar{\mu}^2}$ we can guarantee*

$$\|\theta_{t+1} - \widehat{\theta}\|^2 \leq \left(1 - \frac{1}{80\bar{\kappa}}\right)\|\theta_t - \widehat{\theta}\|^2.$$

*Moreover, let $t_0 \geq 0$ be the first iteration for which $\|\theta_{t_0} - \widehat{\theta}\|^2 < \frac{36\|\mathrm{HT}_s(\nabla f(\widehat{\theta}))\|^2}{\bar{\mu}^2}$. Then, for all $t \geq t_0$, $\|\theta_t - \widehat{\theta}\|^2 \leq \left(1 + \frac{1}{80\bar{\kappa}}\right)\frac{36\|\mathrm{HT}_s(\nabla f(\widehat{\theta}))\|^2}{\bar{\mu}^2}$.*

Theorem 1 implies linear convergence at a rate scaling with $\bar{\kappa}^{-1}$ up to precision $\mathcal{O}\left(\frac{\|\mathrm{HT}_s(\nabla f(\widehat{\theta}))\|^2}{\bar{\mu}^2}\right)$. This result is near equivalent to that in Theorem 3 in Jain et al. (2014), where the RSS constant is assumed to be known, up to a constant factor. Thus, we successfully answer in the affirmative question **(ii)** posed in Section 1. Further, observe that, if $\bar{L}$, $\bar{\mu}$ and $\|\mathrm{HT}_s(\nabla f(\widehat{\theta}))\|^2$ remain constant as the size of the problem grows, the rate remains unchanged and the achievable precision does so too. This implies that for a variety of statistical models the rate and final precision will remain invariant as the problem size increases, as long as the aforementioned quantities do not change. Thus, we also answer in the affirmative question **(iii)** posed in Section 1. We particularize the result to specific statistical models and provide further discussion in Section 4 and Appendix C.

Here, $\widehat{f}$ is a user-defined target value that reflects the desired level of optimization, which can be set above or equal to the statistical accuracy of the problem. Such a relaxation is natural in learning

problems, as it is often counterproductive to optimize to full precision. The continuity and restricted strong convexity of $f$ imply that an $s^*$-sparse vector $\widehat{\theta}$ such that $f(\widehat{\theta}) = \widehat{f}$ always exists.

The additional factor $\frac{1}{80\bar{\kappa}}$ in the final precision (c.f. Theorem 1 when $t \geq t_0$) stems from the expansiveness of the hard thresholding operator. The removal of this factor and support recovery guarantees can be achieved under the signal to noise ratio (SNR) condition (c.f.(3)) and are provided in the following Corollary. Such a condition is widely used in hard thresholding and support recovery studies, as seen in (Shen and Li, 2017a, Prop 2) Bouchot et al. (2016); Shen and Li (2017c); Yuan et al. (2016).

**Corollary 1.** *Under the assumptions stated in Theorem 1, if, further, the SNR condition:*

$$|\widehat{\theta}|_{\min} \geq \frac{7\|\mathrm{HT}_s(\nabla f(\widehat{\theta}))\|}{\bar{\mu}} \tag{3}$$

*holds, for any $t \geq t_0$, where $t_0$ is defined in Theorem 1, the support of $\theta_t$ contains that of $\widehat{\theta}$, and the sequence $\|\theta_t - \widehat{\theta}\|^2$ is non-increasing and upper bounded by $\frac{36\|\mathrm{HT}_s(\nabla f(\widehat{\theta}))\|^2}{\bar{\mu}^2}$.*

Algorithm 1 offers an approach to obtain a target accuracy that we assume known in advance to (1), without requiring precise knowledge of $L, \mu, \tau$. For scenarios in which we only have access to a lower bound on the problem, Algorithm 2 serves as an alternative; in most learning problems, the bound can be simply set to 0, making the method broadly applicable. This adaptive variant of Algorithm 1 builds on the framework of Hazan and Kakade (2019), which reviews gradient descent with Polyak's step size and its double-loop counterpart. By updating the lower bound adaptively in an outer loop, the method ensures that either the accuracy $\mathcal{O}\left(\frac{\|\mathrm{HT}_s(\nabla f(\widehat{\theta}))\|^2}{\bar{\mu}^2}\right)$ is attained or the updated lower bound remains valid.

---

**Algorithm 2** IHT with Adaptive Polyak

---

1: **Input:** Function $f$, a lower bound $\widetilde{f}_1$, and sparsity parameter $s$.
2: **Initialize:** $\theta_0 = 0 \in \mathbb{R}^d$
3: **for** k=1 to $K$ **do**
4:     **for** t = 0 to $T - 1$ **do**
5:         Compute step-size $\gamma_t = \frac{f(\theta_t) - \widetilde{f}_k}{10\|\mathrm{HT}_s(\nabla f(\theta_t))\|^2}$
6:         Update: $\theta_{t+1} = \mathrm{HT}_s\left(\theta_t - \gamma_t g_t\right)$
7:     **end for**
8:     $\bar{\theta}_k = \mathrm{argmin}_{t \leq T} f(\theta_t)$
9:     $\widetilde{f}_{k+1} = \frac{f(\bar{\theta}_k) + \widetilde{f}_k}{2}$
10:    $\theta_0 = \bar{\theta}_k$
11: **end for**
12: **Output:** $\bar{\theta} = \mathrm{argmin}_{k \leq K} f(\bar{\theta}_k)$

---

Let $s \geq s^* > 0$, and define $f^* := \min_{\|\theta\|_0 \leq s^*} f(\theta)$, attained by some $s^*$-sparse vector $\theta^*$.

**Theorem 2.** *Consider the iterates $\{\bar{\theta}_k\}$ generated by Algorithm 2. Assume that the function $f$ fulfills Assumptions 1 and 2. Then for $\varepsilon = (1 + \frac{1}{160\bar{\kappa}})\frac{36(\bar{L}+\bar{\mu})\|\mathrm{HT}_s(\nabla f(\theta^*))\|^2}{\bar{\mu}^2}$, when $\bar{\mu} > 0, s \geq (480\bar{\kappa})^2 s^*$, Algorithm 2 requires at most $\widetilde{T} := \left(1 + \log_2 \frac{2(f(\theta_0) - f(\theta^*))}{\varepsilon}\right)T$ gradient evaluations to achieve $f(\bar{\theta}) - f(\theta^*) \leq \varepsilon$ and $\|\bar{\theta} - \theta^*\|^2 \leq (1 + \frac{1}{160\bar{\kappa}})\frac{36\|\mathrm{HT}_s(\nabla f(\theta^*))\|^2}{\bar{\mu}^2}$. Here $T = \left\lceil \frac{1}{\log(1/(1-1/160\bar{\kappa}))} \log\left(\frac{\bar{\mu}^2\|\theta_0 - \theta^*\|^2}{36(1+1/160\bar{\kappa})\|\mathrm{HT}_s(\nabla f(\theta^*))\|^2}\right)\right\rceil$.*

This theorem focuses on the distance of the iterates to $\theta^*$. The quantity $T$ can be interpreted as the number of iterations required to reach the desired accuracy when applying Algorithm 1 with $\widehat{f} = f^*$. The additional term $\mathcal{O}\left(\frac{f(\theta_0) - f(\theta^*)}{\varepsilon}\right)$ in the definition of $\widetilde{T}$ corresponds to the number of outer iterations needed to obtain a sufficiently tight lower bound for the targeted accuracy. Similar order guarantees are established in Theorems 1 and 2. The proof of Theorem 2 is provided in Appendix B.

# 4 Statistical Guarantees

The results in Section 3 are deterministic in nature and consequently, do not depend on a data generation model. In, contrast, in this section we use Theorem 1 to provide guarantees for specific statistical models. Corollaries 2 and 3 establish the computational-statistical performance guarantees for sparse logistic regression and low-rank matrix regression respectively. We provide guarantees for additional statistical models, including sparse linear regression, in Appendix C.

## 4.1 Logistic Regression

We consider a dataset consisting of observations $z_i = (x_i, y_i)$ for $i = 1, \ldots, n$, where $x_i \in \mathbb{R}^d$ denote the feature vectors, and $y_i \in \{0, 1\}$ denote the corresponding responses. The feature vectors are organized into the design matrix

$$X \triangleq (x_1, \ldots, x_n)^\top \in \mathbb{R}^{n \times d}.$$

We assume that the relationship between $y_i$ and $x_i$ follows the model

$$\Pr(y_i = 1 \mid x_i) = \frac{1}{1 + \exp(-x_i^\top \theta^*)}, \tag{4}$$

where $\theta^*$ is an $s^*$-sparse vector representing the underlying ground truth parameter. The objective function is defined as

$$f(\theta) = \frac{1}{n} \sum_{i=1}^{n} \log \left(1 + \exp(x_i^T \theta)\right) - y_i x_i^T \theta.$$

We assume that each covariate vector $x_i$ is drawn independently from a multivariate normal distribution $\mathcal{N}(0, \Sigma)$, where $\Sigma$ is a non-singular covariance matrix. By invoking Corollary 1 in Yuan and Li (2021), we can establish that the objective function $f(\theta)$ satisfies the RSS and RSC conditions, as formalized in the following lemma.

**Lemma 1.** *Consider the sparse linear logistic regression problem described above. Suppose the covariates $x_i$ are uniformly bounded such that $\|x_i\| \leq 1$ for all $i \in [n]$. Then $f(\theta)$ is $\bar{L}$-smooth with $\bar{L} = 1$. Moreover, with probability at least $1 - e^{-c_0 n}$, the RSC condition holds with curvature parameter $\mu := \frac{1}{2} \exp(-4R) \sigma_{\min}(\Sigma)$ and tolerance $\tau := c_1 \exp(-4R) \zeta(\Sigma) \frac{\log d}{n}$, where $R := \|\theta^*\|$, $\zeta(\Sigma) = \max_{i=1,\ldots,d} \Sigma_{ii}$, and $c_0, c_1 > 0$ are universal constants.*

**Corollary 2.** *Consider the sparse linear logistic regression problem described above. Under the assumptions of Lemma 1, further suppose that the sample size is sufficiently large so that $\bar{\mu} > 0$. Further, assume the design matrix $X \in \mathbb{R}^{n \times d}$ is normalized such that $\|X_j/\sqrt{n}\| \leq C$ for all $j = 1, \ldots, d$. Let $\{\theta_t\}_{t \geq 0}$ be the sequence of iterates produced by Algorithm 1 when applied to the sparse logistic regression problem. Assume the sparsity parameter satisfies $s \geq (240 \bar{\kappa})^2 s^*$, and $\hat{f} = f(\theta^*)$. Then, with probability at least $1 - e^{-c_0 n} - \frac{2}{d}$, the following hold:*

*(i)If $\|\theta_t - \theta^*\|^2 \geq 72 C^2 \frac{s \log d}{n \bar{\mu}^2}$, the iterates exhibit contraction toward $\theta^*$, i.e., $\|\theta_{t+1} - \theta^*\|^2 \leq \left(1 - \frac{1}{80 \bar{\kappa}}\right) \|\theta_t - \theta^*\|^2$. (ii) Let $t_0$ denote the first iteration at which $\|\theta_{t_0} - \theta^*\|^2 < 72 C^2 \frac{s \log d}{n \bar{\mu}^2}$. Then for all $t \geq t_0$, the iterates remain confined in a neighborhood of $\theta^*$: $\|\theta_t - \theta^*\|^2 \leq \left(1 + \frac{1}{80 \bar{\kappa}}\right) 72 C^2 \frac{s \log d}{n \bar{\mu}^2}$.*

*(iii) If $\theta^*$ satisfies the SNR condition (3), then the iterates remain confined in a neighborhood of $\theta^*$ $\|\theta_t - \theta^*\|^2 \leq 72 C^2 \frac{s \log d}{n \bar{\mu}^2}$ for all $t \geq t_0$, and the support of $\theta^*$ is exactly recovered and preserved for all subsequent iterations.*

The proof of Corollary 2 is provided in Appendix A.3.

**Remark 1.** *The assumption $\|x_i\| \leq 1$, $\forall i \in [n]$ is required in Yuan and Li (2021) to provide performance guarantees of HT with a fixed step-size on Logistic Regression. However, we note that this assumption is extremely restrictive in the high-dimensional setting. We provide additional results that do not require $\|x_i\| \leq 1$, $\forall i \in [n]$ in Appendix C.2. Our results in Appendix C.2 further apply to additional GLMs that do not satisfy the RSC condition (2) globally, and are aligned with the results in Loh and Wainwright (2015) both in terms of sample and asymptotic convergence rates. In these cases, convergence at a rate of $(1 - c_0/\bar{\kappa})$ can be guaranteed if the algorithm is suitably initialized. However, a stricter requirement on the sparsity level, specifically $s \geq \mathcal{O}(\bar{\kappa}^4) s^*$,*

*is necessary otherwise, and a rate of $\left(1 - 1/c_1\bar{\kappa}^2\right)$ can be guaranteed. Here $c_0$, and $c_1$ are universal constants.*

The result above and the result in Appendix C.2 match the observed behavior of Sparse Polyak in Fig 1. Namely, we observe that Sparse Polyak indeed achieves rate invariance, as $\frac{d}{n} \to \infty$, the rate and final precision remain constant as long as $\frac{s \log d}{n}$ is left unchanged.

### 4.2 Matrix Regression

Consider the data generation model

$$y_i = \langle X_i, \Theta^* \rangle + \varepsilon_i, \quad \text{for } i = 1, 2 \ldots, n,$$

where $\Theta^* \in \mathbb{R}^{d \times d}$ is a matrix of rank at most $s^*$. We assume, $X_i \in \mathbb{R}^{d \times d}$, $\text{vec}(X_i) \sim \mathcal{N}(0, \Sigma)$ are i.i.d, and $\Sigma \succ 0$. Further, $\varepsilon_i \sim N(0, \sigma^2)$ are i.i.d. and independent of $X_i$. Define $f(\Theta) = \frac{1}{2n} \sum_{i=1}^{n} \left(y_i - \langle X_i, \Theta \rangle\right)^2$.

By invoking (Agarwal et al., 2012, Lemma 7), we establish the RSS and RSC properties of $f(\Theta)$, as formalized in the following lemma.

**Lemma 2.** *Consider the low-rank matrix regression problem described above. Then, with probability at least $1 - e^{-c_0 n}$, $f(\Theta)$ satisfies the RSS and RSC conditions with respect to the Frobenius norm and the nuclear norm. The corresponding parameters are given by:*

$$L = 2\sigma_{\max}(\Sigma), \quad \mu = \frac{1}{2}\sigma_{\min}(\Sigma), \quad \text{and} \quad \tau = c_1 \zeta(\Sigma)\frac{d}{n},$$

*where $\zeta(\Sigma) := \sup_{\|u\|=1, \|v\|=1} \text{Var}(u^\top X_1 v)$, and $c_0, c_1 > 0$ are universal constants.*

To enforce the suitable low-rank structure on the iterates we define

$$\text{PM}_s(W) = \sum_{i=1}^{s} \sigma_i u_i v_i^T,$$

where $\sigma_i, i = 1 \ldots, s$ are the $s$ largest singular values of $W$, and $u_i, v_i$ the corresponding singular vectors. We substitute all instances of the $\text{HT}_s$ operator by $\text{PM}_s$ in Algorithm 1 and 2 (c.f. Appendix B).

**Corollary 3.** *Consider the low rank matrix regression problem described above. Let $\{\Theta_t\}_{t \geq 0}$ be the sequence of iterates generated by Algorithm 1 when applied to a low rank matrix regression problem. Suppose that $\widehat{f} = f(\Theta^*)$, $n$ is sufficiently large such that $\bar{\mu} > 0$, and $s \geq (240\bar{\kappa})^2 s^*$. Then, with probability at least $1 - e^{-c_0 n} - 2e^{-4d}$, the following holds: (i) If $\|\Theta_t - \Theta^*\|_F^2 \geq \frac{7200\sigma^2 \zeta(\Sigma) s d}{n\bar{\mu}^2}$, then the iterates contract relative to $\Theta^*$ as $\|\Theta_{t+1} - \Theta^*\|_F^2 \leq \left(1 - \frac{1}{80\bar{\kappa}}\right)\|\Theta_t - \Theta^*\|_F^2$. (ii) Let $t_0$ be the first iteration for which $\|\Theta_{t_0} - \Theta^*\|_F^2 < \frac{7200\sigma^2 \zeta(\Sigma) s d}{n\bar{\mu}^2}$. Then, for all $t \geq t_0$, the iterates remain in a stable neighborhood around $\Theta^*$, with $\|\Theta_t - \Theta^*\|_F^2 \leq \left(1 + \frac{1}{80\bar{\kappa}}\right)\frac{7200\sigma^2 \zeta(\Sigma) s d}{n\bar{\mu}^2}$.*

The proof of Corollary 3 can be found in Appendix A.4.

**Remark 2.** *The result in Corollary 1 also extends to Algorithm 2. If $\theta^*$ satisfies the SNR condition (3), the iterates of Algorithm 2 recover the support after $\widetilde{T}$ gradient descent steps.*

## 5 Sketch of the Proof for Theorem 1 and Corollary 1

We provide a sketch of the proof of the main results of the paper. We refer the reader to the appendix for the complete proof. The proof of Theorem 1 follows the outline: **(i)** study the behavior of $\|\theta_{t+1} - \widehat{\theta}\|^2$ given $\gamma_t$ and $\|\theta_t - \widehat{\theta}\|$, **(ii)** establish that under the assumption that $\gamma_t$ is sufficiently large the expansive effect of the Hard Thresholding operator can be offset by the contractive effect of the gradient update, **(iii)** show that $\gamma_t$ is sufficiently large until we reach optimal statistical precision. Both to finalize the proof of Theorem 1 and to establish Corollary 1: **(iv)** the iterates remain confined within a neighborhood of $\theta^*$, and, given that the SNR condition holds, the support can be identified, providing further benefits to the algorithm's performance. We elaborate on points **(i)** through **(iv)**.

**(i)** To understand the dynamics of $\|\theta_t - \widehat{\theta}\|^2$ we analyze the combined effect of gradient descent under the RSC and RSS and the Hard Thresholding operator.

For this we suitably apply Lemma 1 from Jain et al. (2014), exploit the RSC, and the properties of $\theta_{t+1}$ in relation to $\text{HT}_s$ to yield:

$$\|\theta_{t+1} - \theta^*\|^2 \leq \left(1 + \sqrt{\frac{s^*}{s}}\right)^2 \left((1 - \bar{\mu}\gamma_t)\|\theta_t - \theta^*\|^2 \right. \tag{5a}$$

$$\left. -2\gamma_t \left(f(\theta_t) - f(\theta^*)\right) + 10\gamma_t^2 \|\text{HT}_s(\nabla f(\theta_t))\|^2\right). \tag{5b}$$

Setting $\gamma_t = \frac{f(\theta_t) - f(\theta^*)}{5\|\text{HT}_s(\nabla f(\theta_t))\|^2}$ makes (5b) nonpositive and yields a choice that is invariant with $d$. The use of $\|\text{HT}_s(\nabla f(\theta_t))\|^2$ in $\gamma_t$ and the RSS are critical to avoiding a step-size that scales with the Lipschitz smoothness constant (which scales as $\mathcal{O}(d)$ in the high-dimensional setting). Setting $\gamma_t = 1/(40\bar{L})$ in (5) recovers the result in Jain et al. (2014), which establishes that IHT requires at most $\mathcal{O}(\bar{\kappa}^{-1}\log(1/\varepsilon))$ to be within $\varepsilon-$accuracy of near optimal statistical precision.

**(ii)** To achieve the optimal linear rate under our choice of step size, we require that

$$\left(1 + \sqrt{\frac{s^*}{s}}\right)^2 (1 - \bar{\mu}\gamma_t) \leq 1 - c_0\frac{\bar{\mu}}{\bar{L}}. \tag{6}$$

Sufficient conditions for (6) are $s \geq c_1\bar{\kappa}^2 s^*$ and $\gamma_t \geq \frac{c_2}{\bar{L}}$ for some universal constants $c_0$, $c_1$ and $c_2$. However, if $\gamma_t$ were to scale with the Lipschitz smoothness constant, i.e. with $d^{-1}$ we would require $s \geq d^2 s^*$ to establish linear convergence. Observe that this requirement can not be fulfilled as $s \leq d$.

**(iii)** We exploit the RSS and RSC to show that $\gamma_t \geq \frac{1}{40\bar{L}}$ when $\|\theta_t - \widehat{\theta}\|^2 \geq \frac{36\|\nabla f(\widehat{\theta})\|_s^2}{\bar{\mu}^2}$. If, further, $s \geq (240\bar{\kappa})^2 s^*$, (6) holds with $c_0 = 1/160$, yielding $\|\theta_{t+1} - \widehat{\theta}\|^2 \leq \left(1 - c_0\bar{\kappa}^{-1}\right)\|\theta_t - \widehat{\theta}\|^2$.

The condition $\|\theta_t - \widehat{\theta}\|^2 \geq \frac{36\|\text{HT}_s(\nabla f(\widehat{\theta}))\|^2}{\bar{\mu}^2}$ stems from (1) being a constrained problem. If the Polyak step-size had been left unaltered, additional regularity conditions are required to establish convergence in the constrained case. As established in Theorem 3 in Polyak (1969), one such condition is $\frac{f(\theta_t) - f(\widehat{\theta})}{\|\theta_t - \widehat{\theta}\|} \geq c$ for some $c > 0$ and any $\theta_t \in \mathbb{R}^d$, which does not uniformly in high-dimensional M-estimation. For some GLMs, we can show the condition holds locally and exploit this fact to provide more general results in Appendix C.2.

**(iv)** From **(i)**-**(iii)** it follows that there exists $t_0$ at which $\|\theta_{t_0} - \widehat{\theta}\|^2 \leq \frac{36\|\text{HT}_s(\nabla f(\widehat{\theta}))\|^2}{\bar{\mu}^2}$. Since the potential expansion is at most $\left(1 + \frac{1}{80\bar{\kappa}}\right)$ Theorem 1 follows. If $\widehat{\theta}$ satisfies (3), we show that the inequality $\|\theta_{t_0} - \widehat{\theta}\|^2 \leq \frac{36\|\text{HT}_s(\nabla f(\widehat{\theta}))\|^2}{\bar{\mu}^2}$ guarantees that $\widehat{\mathcal{S}} \subset \mathcal{S}_{t_0}$. From here, we establish that, this results in two possible scenarios: **(a)** $\gamma_t < \frac{1}{40\bar{L}}$, implying $\widehat{\mathcal{S}} \subset \mathcal{S}_{t_0+1}$, ensuring that $\|[\tilde{\theta}_{t_0+1} - \widehat{\theta}]_{\widehat{\mathcal{S}}_{t+1}}\|^2 = \|\tilde{\theta}_{t_0+1} - \widehat{\theta}\|^2$, and eliminating the expansion term in (5); or, **(b)** $\gamma_t \geq \frac{1}{40\bar{L}}$, and (6) holds. Based on **(a)** and **(b)**, Corollary 1 is established by induction.

## 6 Numerical experiments

We first consider sparse linear regression and sparse logistic regression on synthetic data. This is done to illustrate the algorithm's performance as the size of the problem grows while the problem conditioning is kept the same. In all scenarios that rely on synthetic data, we set $d \in \{5000, 10000, 20000\}$, and $s^* = 300$. The design matrix $X \in \mathbb{R}^{d \times n}$ is generated to reflect a time-series structure with a correlation parameter $\omega = 0.5$. We set the sample size $n$ according to $n = \lceil \alpha s \log d \rceil$, where $\alpha > 0$ is a constant. In this section, we use $\alpha = 5$. For each column index $j \in \{1, \ldots, n\}$, we generate a sequence of i.i.d. standard normal variables $\varepsilon_1, \ldots, \varepsilon_{d-1}$, and construct $x_{1,j} = \varepsilon_1/\sqrt{1 - \omega^2}$. The subsequent entries are generated recursively as $x_{t+1,j} = \omega x_{t,j} + \varepsilon_t$ for $t \in \{1, \ldots, d-1\}$, where $\varepsilon_t \sim \mathcal{N}(0, 1)$. The true parameter $\theta^*$ is created by sampling each entry from $\mathcal{N}(0, 1)$, and assigning nonzero values to $s^*$ randomly chosen entries.

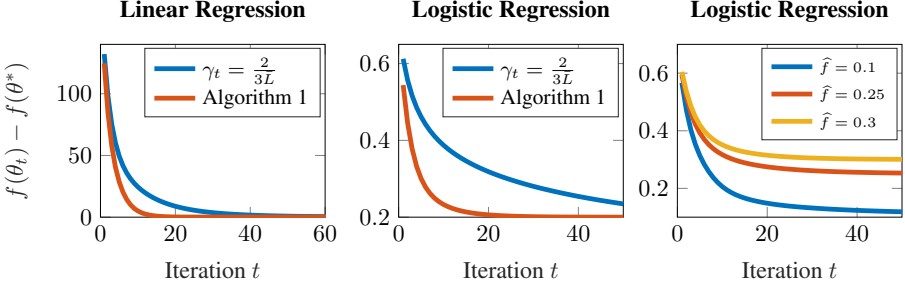

Figure 2: **Left and center:** IHT with $\frac{2}{3\bar{L}}$ (blue) vs. Algorithm 1 (red) on linear and logistic regression respectively. **Right:** Choice of $\widehat{f}$ on Algorithm 1. In all scenarios $\alpha = 5$, $d = 5000$ and $s = 700$.

In the case of linear regression, each sample $i$ (where $i \in \{1, \ldots, n\}$) is generated according to the model:
$$y_i = x_i^T \theta^* + w_i, \quad w_i \sim \mathcal{N}(0, 0.25).$$
For logistic regression, the relationship between $y_i$ and $x_i$ follows the model (4).

**(i) Comparison to fixed step-size:** In Figure 6 we present a comparison between IHT with a fixed step size (blue) and the adaptive step size used in Algorithm 1 (red) when solving linear regression (left panel) and logistic regression (center panel) respectively. When working with a constant step-size, we set the step size to $\frac{2}{3\bar{L}}$ following Jain et al. (2014) for both linear and logistic regression. $\bar{L}$ for linear regression can be upper bounded as $\lambda_{\max}(\Sigma)(3 + \frac{2(2s+s^*)}{s\alpha})$ (Loh and Wainwright, 2015, Appendix D.1), whereas $\bar{L}$ for logistic regression is one fourth of that of linear regression. In both settings, $\lambda_{\max}(\Sigma) \leq \frac{2}{(1-\omega)^2(1+\omega)}$ Agarwal et al. (2012). Although both step size strategies share the same theoretical guarantees, we observe that the adaptive step size speeds-up convergence. This advantage arises because an adaptive step size can adapt to the local curvature and therefore be significantly more aggressive, allowing for faster progress towards the solution. As expected, this effect is more pronounced when solving logistic regression where the functions' curvature will depend on the point.

**(ii) Rate invariance:** In Figure 1 (c.f. Section 1), we compare the classical Polyak step size with the step size employed in Algorithm 1 across different problem dimensions $d$, while maintaining $\alpha = 5$. The solid line represents the performance of Algorithm 1, whereas the dashed line corresponds to the classical Polyak step size. This demonstrates that when the condition number of $\Sigma$ remains unchanged, the complexity of the method remains almost identical under our chosen step size. In contrast, Polyak's step size leads to an increased number of iterations, even if the achievable statistical precision and $(\lambda_{\max}(\Sigma)/\lambda_{\min}(\Sigma))$ remain the same.

**(iii) Choice of $\widehat{f}$ :** Finally, Figure 6 (right) highlights the impact of the choice of $\widehat{f}$. The results confirm that $\widehat{f}$ determines the best achievable accuracy. Additionally, from the formulation of $\gamma_t$, we observe that $\widehat{f}$ directly influences the step size magnitude, thus is impacting the convergence rate.

Experiments on real world data are provided in Appendix D. All experiments were conducted on a laptop equipped with 16 GB of RAM and a 12th Gen Intel Core i5-12500H 3.10 GHz CPU.

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

# Appendix

## A  Main Theorems

In this section we provide the formal proof to the statements included in the main body of the paper. To formally establish Theorem 1 we build on (Jain et al., 2014, Lemma 1) and Lemmas 4-6. (Jain et al., 2014, Lemma 1) allows us to control the expansive properties of the Hard Thresholding operator, whereas Lemma 4 allows us to establish (5) (c.f. Section 5). Further, in Lemma 5 we establish consequences of the RSS (Assumption 2) that are instrumental in establishing a lower bound on the step-size $\gamma_t$. We lower bound $\gamma_t$ in Lemma 6 and establish conditions under which this lower bound holds. We combine these results in the proof of Theorem 1 in Appendix A.1, followed by a formal proof of the corollaries included in the main body of the paper in the remaining sections of Appendix A.

For simplicity, we let $\widehat{\mathcal{S}}_t = \mathcal{S}_t \cup \widehat{\mathcal{S}}$. Also, we let $g_t = \nabla f(\theta_t)$, and $\widehat{g} = \nabla f(\widehat{\theta})$. When discussing specific statistical models we denote by $\theta^*$ the ground truth, $g^* = \nabla f(\theta^*)$ and $\mathcal{S}^* = \text{supp}(\theta^*)$. For any index set $\mathcal{S}$ and vector $\theta \in \mathbb{R}^d$, we define $[\theta]_{\mathcal{S}}$ as the vector that retains the entries indexed by $\mathcal{S}$, while setting all other entries to zero.

We include the following fundamental lemma, which plays a key role in our analysis. It corresponds to Lemma 1 in Jain et al. (2014), and is presented here for completeness.

**Lemma 3.** *For any index set $I$, any $z \in \mathbb{R}^I$, let $\theta = \text{HT}_s(z)$. Then for any $\widehat{\theta} \in \mathbb{R}^I$ such that $\|\widehat{\theta}\|_0 \leq s^*$, we have*

$$\|\theta - z\|^2 \leq \frac{|I| - s}{|I| - s^*}\|\widehat{\theta} - z\|^2.$$

**Lemma 4.** *Let $\widehat{\theta}$ be any $s^*$-sparse vector, and $\theta_t$ be any $s$-sparse vector. Assume that the function $f$ fulfills Assumption 1 with $\bar{\mu} = \mu - 3\tau s > 0$. Let $\theta_{t+1} := \text{HT}_s\left(\theta_t - \gamma_t g_t\right)$, and $\widetilde{\theta}_{t+1} := \theta_t - \gamma_t g_t$. For any $\gamma_t \leq \frac{f(\theta_t) - f(\widehat{\theta})}{5\|\text{HT}_s(\widehat{g})\|^2}$ we have*

$$\|\theta_{t+1} - \widehat{\theta}\|^2 \leq \left(1 + \sqrt{\frac{s^*}{s}}\right)^2 \left\|\left[\widetilde{\theta}_{t+1}\right]_{\widehat{\mathcal{S}}_{t+1}} - \widehat{\theta}\right\|^2$$

$$\leq \left(1 + \sqrt{\frac{s^*}{s}}\right)^2 (1 - \bar{\mu}\gamma_t)\|\theta_t - \widehat{\theta}\|^2.$$

*Proof.* For the first inequality,

$$\|\theta_{t+1} - \widehat{\theta}\| \overset{(i)}{\leq} \|\theta_{t+1} - [\widetilde{\theta}_{t+1}]_{\widehat{\mathcal{S}}_{t+1}}\| + \|[\widetilde{\theta}_{t+1}]_{\widehat{\mathcal{S}}_{t+1}} - \widehat{\theta}\| \overset{(ii)}{\leq} \left(1 + \sqrt{\frac{s^*}{s}}\right) \|[\widetilde{\theta}_{t+1}]_{\widehat{\mathcal{S}}_{t+1}} - \widehat{\theta}\|,$$

where in $(i)$ we use the triangle inequality and in $(ii)$ we used Lemma 3.

For the second inequality, we consider the expansion

$$\|[\tilde{\theta}_{t+1}]_{\widehat{\mathcal{S}}_{t+1}} - \widehat{\theta}\|^2 = \|[\tilde{\theta}_{t+1} - \widehat{\theta}]_{\widehat{\mathcal{S}}_{t+1}}\|^2$$
$$= \|[\theta_t - \widehat{\theta}]_{\widehat{\mathcal{S}}_{t+1}}\|^2 - 2\gamma_t\langle\theta_t - \widehat{\theta}, [g_t]_{\widehat{\mathcal{S}}_{t+1}}\rangle + \gamma_t^2\|[g_t]_{\widehat{\mathcal{S}}_{t+1}}\|^2,$$

where

$$-2\gamma_t\langle\theta_t - \widehat{\theta}, [g_t]_{\widehat{\mathcal{S}}_{t+1}}\rangle = -2\gamma_t\langle\theta_t - \widehat{\theta}, g_t\rangle + 2\gamma_t\langle\theta_t - \widehat{\theta}, [g_t]_{\mathcal{S}_t\setminus\widehat{\mathcal{S}}_{t+1}}\rangle,$$

and consequently

$$\left\|[\tilde{\theta}_{t+1} - \widehat{\theta}]_{\widehat{\mathcal{S}}_{t+1}}\right\|^2 \le \|\theta_t - \widehat{\theta}\|^2 - 2\gamma_t\langle\theta_t - \widehat{\theta}, g_t - [g_t]_{\mathcal{S}_t\setminus\widehat{\mathcal{S}}_{t+1}}\rangle + \gamma_t^2\left\|[g_t]_{\widehat{\mathcal{S}}_{t+1}}\right\|^2$$

Using the RSC yields

$$\|[\tilde{\theta}_{t+1} - \widehat{\theta}]_{\widehat{\mathcal{S}}_{t+1}}\|^2 \le (1 - \bar{\mu}\gamma_t)\|\theta_t - \widehat{\theta}\|^2 - 2\gamma_t(f(\theta_t) - f(\widehat{\theta})) + \gamma_t^2\|\mathrm{HT}_{s+s^*}(g_t)\|^2$$
$$+ 2\gamma_t\langle\theta_t - \widehat{\theta}, [g_t]_{\mathcal{S}_t\setminus\widehat{\mathcal{S}}_{t+1}}\rangle. \tag{7}$$

To obtain (5) we must upper bound the inner product in (7), for which we have:

$$\langle[\theta_t]_{\mathcal{S}_t\setminus\widehat{\mathcal{S}}_{t+1}}, \gamma_t[g_t]_{\mathcal{S}_t\setminus\widehat{\mathcal{S}}_{t+1}}\rangle = \langle[\theta_t]_{\mathcal{S}_t\setminus\widehat{\mathcal{S}}_{t+1}}, \gamma_t[g_t]_{\mathcal{S}_t\setminus\widehat{\mathcal{S}}_{t+1}}\rangle$$
$$= \langle[\theta_t]_{\mathcal{S}_t\setminus\widehat{\mathcal{S}}_{t+1}} - \gamma_t[g_t]_{\mathcal{S}_t\setminus\widehat{\mathcal{S}}_{t+1}} + \gamma_t[g_t]_{\mathcal{S}_t\setminus\widehat{\mathcal{S}}_{t+1}}, \gamma_t[g_t]_{\mathcal{S}_t\setminus\widehat{\mathcal{S}}_{t+1}}\rangle$$
$$\le \|[\theta_t]_{\mathcal{S}_t\setminus\widehat{\mathcal{S}}_{t+1}} - \gamma_t[g_t]_{\mathcal{S}_t\setminus\widehat{\mathcal{S}}_{t+1}}\|\|\gamma_t[g_t]_{\mathcal{S}_t\setminus\widehat{\mathcal{S}}_{t+1}}\| + \|\gamma_t[g_t]_{\mathcal{S}_t\setminus\widehat{\mathcal{S}}_{t+1}}\|^2.$$

Then,

$$\|[\theta_t - \gamma_t g_t]_{\mathcal{S}_t\setminus\widehat{\mathcal{S}}_{t+1}}\| \overset{(i)}{\le} \|[\theta_t - \gamma_t g_t]_{\mathcal{S}_t\setminus\mathcal{S}_{t+1}}\| \overset{(ii)}{\le} \|[\theta_t - \gamma_t g_t]_{\mathcal{S}_{t+1}\setminus\mathcal{S}_t}\| = \|[\gamma_t g_t]_{\mathcal{S}_{t+1}\setminus\mathcal{S}_t}\|,$$

where in $(i)$ we use that $\mathcal{S}_t \setminus \widehat{\mathcal{S}}_{t+1} \subseteq \mathcal{S}_t \setminus \mathcal{S}_{t+1}$, and in $(ii)$ we exploit that $|\mathcal{S}_t \setminus \mathcal{S}_{t+1}| = |\mathcal{S}_{t+1} \setminus \mathcal{S}_t|$ and that $\mathcal{S}_{t+1}$ contains the indexes of the $s$ largest elements of $\theta_t - \gamma_t g_t$. Thus, we obtain the overall upper bound

$$2\gamma_t\langle\theta_t - \widehat{\theta}, [g_t]_{\mathcal{S}_t\setminus\widehat{\mathcal{S}}_{t+1}}\rangle \le 2\gamma_t^2\|[g_t]_{\mathcal{S}_t\setminus\widehat{\mathcal{S}}_{t+1}}\|\|[g_t]_{\mathcal{S}_{t+1}\setminus\mathcal{S}_t}\| + 2\gamma_t^2\|[g_t]_{\mathcal{S}_t\setminus\widehat{\mathcal{S}}_{t+1}}\|^2,$$

which together with (7) yields

$$\|[\tilde{\theta}_{t+1} - \widehat{\theta}]_{\widehat{\mathcal{S}}_t}\|^2 \le (1 - \bar{\mu}\gamma_t)\|\theta_t - \widehat{\theta}\|^2 - 2\gamma_t(f(\theta_t) - f(\widehat{\theta})) + 5\gamma_t^2\|\mathrm{HT}_{2s}(g_t)\|^2. \tag{8}$$

Given the upperbound on $\gamma_t$, the two right most terms together are negative, and we thus the proof is complete. $\qquad\square$

Observe that in the proof of Lemma 4 we use the iterate $\tilde{\theta}_{t+1}$ to treat the effect of the hard thresholding operator and gradient descent separately. We then restrict to the support $\widehat{\mathcal{S}}_{t+1}$ to avoid the scaling of any bound with the ambient dimension $d$.

**Lemma 5.** *(RSS-gradient bound) Assume that f fulfills Assumptions 1 where $\bar{\mu} = \mu - 3\tau s > 0$, and Assumption 2. Then, for any pair $x, y$ of $s-$sparse vectors there holds*

$$\frac{1}{2(L + 3\tau s)}\|\mathrm{HT}_s(\nabla f(x) - \nabla f(y))\|^2 \le f(y) - f(x) - \langle\nabla f(x), y - x\rangle.$$

*Proof.* We define

$$\phi(t) = f(t) - \langle\nabla f(x), t - x\rangle.$$

From its formulation, we know $\phi(t)$ inherits the RSS and RSC property of $f$.

As a result, for any $2s$-sparse vector $z$, we have

$$\phi(x) \le \phi(z) - \langle 0, z - x\rangle - \frac{\bar{\alpha}}{2}\|x - z\|^2$$
$$= \phi(z) - \frac{\bar{\alpha}}{2}\|x - z\|^2$$
$$\le \phi(z)$$
$$\le \phi(y) + \langle\nabla\phi(y), z - y\rangle + \frac{L + 3\tau s}{2}\|z - y\|^2.$$

Set $z = y - \frac{1}{L+3\tau s}\mathrm{HT}_s\left(\nabla\phi(y)\right)$, the inequality above indicates

$$\phi(x) \leq \phi(y) - \frac{1}{2(L+3\tau s)}\|\mathrm{HT}_s\left(\nabla\phi(y)\right)\|^2,$$

which is equivalent to

$$f(x) + \langle\nabla f(x), y - x\rangle + \frac{1}{2(L+3\tau s)}\|\mathrm{HT}_s(\nabla f(y) - \nabla f(x))\|^2 \leq f(y).$$

$\square$

**Lemma 6.** *Consider the iterates $\{\theta_t\}_{t\geq1}$ generated by Algorithm 1 to solve* (1). *Assume that $f$ fulfills Assumptions 1 where $\bar{\mu} = \mu - 3\tau s > 0$, and 2. Further, denote by $\widehat{\theta}$ an arbitrary $s^*-$sparse vector for which $f(\widehat{\theta})$ is known and desirable. Then, the step-size*

$$\gamma_t = \frac{f(\theta_t) - f(\widehat{\theta})}{5\|\mathrm{HT}_s(g_t)\|^2} \geq \frac{1}{40\bar{L}}$$

*for each $t \geq 0$ for which*

$$\|\theta_t - \widehat{\theta}\|^2 \geq \frac{18\|\mathrm{HT}_{s+s^*}(\widehat{g})\|^2}{\bar{\mu}^2}$$

$$f(\theta_t) > f(\widehat{\theta}).$$

*Proof.* Assume that $f(\theta_t) - f(\widehat{\theta}) > 0$, then

$$\gamma_t \geq \frac{f(\theta_t) - f(\widehat{\theta})}{10\|\mathrm{HT}_s(g_t - \widehat{g})\|^2 + 10\|\mathrm{HT}_s(\widehat{g})\|^2}.$$

Given that $f$ fulfills Assumption 2 we may invoke Lemma 5 yielding the bound

$$\gamma_t \geq \frac{f(\theta_t) - f(\widehat{\theta})}{20\bar{L}(f(\theta_t) - f(\widehat{\theta}) - \langle\widehat{g}, \theta_t - \widehat{\theta}\rangle) + 10\|\mathrm{HT}_s(\widehat{g})\|^2}.$$

If

$$10\left(2\bar{L}\langle\widehat{g}, \widehat{\theta} - \theta_t\rangle + \|\mathrm{HT}_s(\widehat{g})\|^2\right) \leq 20\bar{L}(f(\theta_t) - f(\widehat{\theta})) \tag{9}$$

we can guarantee that

$$\gamma_t \geq \frac{1}{40\bar{L}}.$$

Rearranging (9) we have that the condition can be equivalently written as

$$\|\mathrm{HT}_s(\widehat{g})\|^2 \leq 2\bar{L}\left(f(\theta_t) - f(\widehat{\theta}) + \langle\widehat{g}, \theta_t - \widehat{\theta}\rangle\right).$$

Invoking the RSC, a sufficient condition for the above is

$$\|\mathrm{HT}_s(\widehat{g})\|^2 \leq 2\bar{L}(\frac{\bar{\mu}}{2}\|\theta_t - \widehat{\theta}\|^2 + 2\langle\widehat{g}, \theta_t - \widehat{\theta}\rangle),$$

which can be guaranteed as long as

$$\|\mathrm{HT}_s(\widehat{g})\|^2 \leq 2\bar{L}(\frac{\bar{\mu}}{2}\|\theta_t - \widehat{\theta}\|^2 - 2\|\mathrm{HT}_{s+s^*}(\widehat{g})\|\|\theta_t - \widehat{\theta}\|)$$

$$= 2\bar{L}\left(\frac{\bar{\mu}}{2}\|\theta_t - \widehat{\theta}\| - 2\|\mathrm{HT}_{s+s^*}(\widehat{g})\|\right)\|\theta_t - \widehat{\theta}\|.$$

To guarantee that the above holds it is sufficient to request that $\|\theta_t - \widehat{\theta}\| \geq \frac{106}{25\bar{\mu}}\|\mathrm{HT}_{s+s^*}(\widehat{g})\|$, and thus the result follows.

$\square$

## A.1 Proof of Theorem 1

As a consequence of Lemma 6 we distinguish three cases for any $t$ : (i) $\|\theta_t - \widehat{\theta}\|^2 \geq \frac{36\|\mathrm{HT}_s(\widehat{g})\|^2}{\bar{\mu}^2}$ and $f(\theta_t) - \widehat{f} > 0$, (ii) $\|\theta_t - \widehat{\theta}\|^2 < \frac{36\|\mathrm{HT}_s(\widehat{g})\|^2}{\bar{\mu}^2}$ and $f(\theta_t) - \widehat{f} > 0$, or (iii) $f(\theta_t) - \widehat{f} \leq 0$.

In case (iii) no progress is made, i.e. $\theta_{t+1} = \theta_t$ and by the RSC there holds

$$\frac{\bar{\mu}}{2}\|\theta_t - \widehat{\theta}\|^2 \leq \|\theta_t - \widehat{\theta}\|\|\mathrm{HT}_{s+s^*}(\widehat{g})\|$$

and thus

$$\|\theta_{t+1} - \widehat{\theta}\|^2 = \|\theta_t - \widehat{\theta}\|^2 < \frac{36\|\mathrm{HT}_s(\widehat{g})\|^2}{\bar{\mu}^2}.$$

Further, from the above it follows that if $\|\theta_t - \widehat{\theta}\|^2 \geq \frac{36\|\mathrm{HT}_s(\widehat{g})\|^2}{\bar{\mu}^2}$ we can guarantee that $\gamma_t > 0$.

For case (i) we begin by invoking Lemma 4, which guarantees that

$$\|\theta_{t+1} - \widehat{\theta}\|^2 \leq \left(1 + 3\sqrt{\frac{s^*}{s}}\right)^2 (1 - \bar{\mu}\gamma_t)\|\theta_t - \widehat{\theta}\|^2.$$

Using Lemma 6 we can guarantee a lower bound on the step-size $\gamma_t \geq \frac{1}{40\bar{L}}$. Further, under our assumption on $s$, namely, $s \geq (240\bar{\kappa})^2 s^*$, we can bound the contraction factor:

$$\left(1 + \sqrt{\frac{s^*}{s}}\right)^2 (1 - \bar{\mu}\gamma_t) \leq \left(1 + \sqrt{\frac{s^*}{s}}\right)(1 - \bar{\mu}\gamma_t) \leq \left(1 + \frac{1}{80\bar{\kappa}}\right)\left(1 - \frac{1}{40\bar{\kappa}}\right) \leq 1 - \frac{1}{80\bar{\kappa}},$$

and therefore,

$$\|\theta_{t+1} - \widehat{\theta}\|^2 \leq \left(1 - \frac{1}{80\bar{\kappa}}\right)\|\theta_t - \widehat{\theta}\|^2.$$

Thus, the first part of the theorem's statement follows, i.e. when $\|\theta_t - \widehat{\theta}\|$ is sufficiently large, we can guarantee that $\theta_{t+1}$ will approach $\widehat{\theta}$. We are now left with establishing the veracity of the second statement. For this, let $t_0$ be the time defined in the theorem's statement. Then, we are under case (ii) or case (iii). If we are under case (iii) there is nothing left to proof. If we are in case (ii), there are two further cases: (a) the iterates remain confined within a ball of radius $6\|\mathrm{HT}_s(\widehat{g})\|^2/\bar{\mu}^2$, i.e.

$$\forall t \geq t_0, \ \|\theta_t - \widehat{\theta}\|^2 < \frac{36\|\mathrm{HT}_s(\widehat{g})\|^2}{\bar{\mu}^2},$$

and the theorem's second statement is therefore true, or (b) there exists a time $t_1 > t_0$ at which for the first time

$$\frac{36\|\mathrm{HT}_s(\widehat{g})\|^2}{\bar{\mu}^2} \leq \|\theta_{t_1} - \widehat{\theta}\|^2.$$

By Lemma 4 and by definition of $t_1$ we have

$$\|\theta_{t_1} - \widehat{\theta}\|^2 \leq \left(1 + \frac{1}{80\bar{\kappa}}\right)(1 - \bar{\mu}\gamma_t)\frac{36\|\mathrm{HT}_s(\widehat{g})\|^2}{\bar{\mu}^2} < \left(1 + \frac{1}{80\bar{\kappa}}\right)\frac{36\|\mathrm{HT}_s(\widehat{g})\|^2}{\bar{\mu}^2}.$$

This implies, we find ourselves again in case (i). Observe that going forward, no iterate can escape the above ball and therefore the second statement of the theorem holds.

## A.2 Proof of Corollary 1

If (3) holds for $\widehat{\theta}$, for any $\theta_t$ fulfilling $\|\theta_t - \widehat{\theta}\|^2 \leq \frac{36\|\mathrm{HT}_s(\widehat{g})\|^2}{\bar{\mu}^2}$ there holds

$$\min_{i \in \widehat{\mathcal{S}}} \ |[\theta_t]_i| \geq \frac{7\|\mathrm{HT}_s(\widehat{g})\|}{\bar{\mu}} - \frac{6\|\mathrm{HT}_s(\widehat{g})\|}{\bar{\mu}} > 0. \tag{10}$$

This can be established by contradiction, i.e. if the condition above is violated, either $\|\theta_t - \widehat{\theta}\|^2 > \frac{36\|\mathrm{HT}_s(\widehat{g})\|^2}{\bar{\mu}^2}$ or (3) can not hold. Consequently, $\widehat{\mathcal{S}} \subset \mathcal{S}^t$. Now we consider the iterate $\theta_{t+1}$. Note that $\widehat{\mathcal{S}} \subset \mathcal{S}^{t+1}$, if

$$\gamma_t \|g_t\|_\infty < \frac{\|\mathrm{HT}_s(\widehat{g})\|}{2\bar{\mu}}. \tag{11}$$

To see this note that for any $i \notin \mathcal{S}_t$ we have

$$|[\theta_t]_i - \gamma_t [g_t]_i| = \gamma_t |[g_t]_i| < \frac{\|\mathrm{HT}_s(\widehat{g})\|}{2\bar{\mu}},$$

and for any $i \in \widehat{\mathcal{S}}$, given that $\|\theta_t - \widehat{\theta}\| \leq \frac{6\|\mathrm{HT}_s(\widehat{g})\|}{\bar{\mu}}$, we have

$$|[\theta_t]_i - \gamma_t [g_t]_i| > \frac{7\|\mathrm{HT}_s(\widehat{g})\|}{\bar{\mu}} - \frac{6\|\mathrm{HT}_s(\widehat{g})\|}{2\bar{\mu}} - \frac{\|\mathrm{HT}_s(\widehat{g})\|}{2\bar{\mu}} = \frac{\|\mathrm{HT}_s(\widehat{g})\|}{2\bar{\mu}}.$$

Because the Hard Thresholding operator selects the $s$ largest components of $\theta_{t+1}$, in the selection of the elements that should go into $\mathcal{S}^{t+1}$ the operator will not deselect elements from $\widehat{\mathcal{S}}$ in benefit of any outside of $\mathcal{S}^t$. Thus, $\widehat{\mathcal{S}} \subset \mathcal{S}^{t+1}$.

We now find conditions on $\gamma_t$ under which (11) can be guaranteed. Observe that

$$\gamma_t \|g_t\|_\infty \leq \gamma_t \left( \|\widehat{g}\|_\infty + \|g_t - \widehat{g}\|_\infty \right)$$
$$\overset{(i)}{\leq} \gamma_t \left( \|\widehat{g}\|_\infty + \bar{L}\|\theta_t - \widehat{\theta}\| \right) \overset{(ii)}{\leq} \gamma_t \left( \|\widehat{g}\|_\infty + \frac{6\bar{L}}{\bar{\mu}}\|\mathrm{HT}_s(\widehat{g})\| \right),$$

where $(i)$ follows from Lemma 5, and $(ii)$ follows from the assumption that $\|\theta_t - \widehat{\theta}\|^2 \leq \frac{36\|\mathrm{HT}_s(\widehat{g})\|^2}{\bar{\mu}}$. A sufficient condition for (11) to hold is thus given by

$$\gamma_t < \frac{1}{2\bar{\mu} + 12\bar{L}}.$$

As a result, when $\gamma_t < \frac{1}{2\bar{\mu}+12\bar{L}}$, we have $\widehat{\mathcal{S}} \subset \mathcal{S}^{t+1}$, and thus $\|\theta_{t+1} - \widehat{\theta}\|^2 \leq \|\theta_t - \widehat{\theta}\|^2$ by $\|\theta_{t+1} - \widehat{\theta}\|^2 = \|[\theta_{t+1} - \widehat{\theta}]_{\widehat{\mathcal{S}}_{t+1}}\|^2 \leq (1 - \bar{\mu}\gamma_t)\|\theta_t - \widehat{\theta}\|^2$. Otherwise, $\gamma_t \geq \frac{1}{2\bar{\mu}+12\bar{L}} > \frac{1}{40\bar{L}}$, we still have $\|\theta_{t+1} - \widehat{\theta}\|^2 \leq \|\theta_t - \widehat{\theta}\|^2$ by Lemma 4. Thus, the proof is completed by induction.

### A.3 Proof of Corollary 2

By invoking Lemma 1, we can guarantee that $f$ fulfills the RSC and RSM with probability at least $1 - e^{-c_0 n}$. The function $f$ is convex by construction, and assuming that Algorithm 1 is provided suitable parameters $\widehat{f}$ and $s$ as stipulated by the Corollary, we may invoke Theorem 1. Thus, with probability at least $1 - e^{-c_0 n}$, the iterates satisfy a contractive relation for $\|\theta_t - \theta^*\|^2$ until the point where $\|\theta_t - \theta^*\|^2 < \frac{36\|\mathrm{HT}_s(g^*)\|^2}{\bar{\mu}^2}$.

To complete the proof, it remains to establish that, with probability at least $1 - \frac{2}{d}$, it holds that

$$\frac{36\|\mathrm{HT}_s(g^*)\|^2}{\bar{\mu}^2} \leq 72C^2 \frac{s \log d}{n\bar{\mu}^2}. \tag{12}$$

The proof of this claim essentially follows the arguments in (Wainwright, 2019, Example 7.14); for completeness, we include the full details here.

Define $\sigma(x) = \frac{1}{1+\exp(-x)}$. The gradient of $f$ evaluated at $\theta^*$ can be expressed as

$$g^* = \frac{1}{n} \sum_{i=1}^n \left( \sigma(x_i^\top \theta^*) - y_i \right) x_i.$$

Recall the relation between $x_i$ and $y_i$ is governed by the model

$$\Pr(y_i = 1 \mid x_i) = \frac{1}{1 + \exp(-x_i^\top \theta^*)}.$$

Under this model, it follows that each term $\sigma(x_i^\top \theta^*) - y_i$ is a zero-mean sub-Gaussian random variable with sub-Gaussian parameter $\sigma^2 = \frac{1}{4}$.

Thus, $\|g^*\|_\infty$ is the maximum of $d$ independent zero-mean sub-Gaussian random variables, each with variance proxy at most $\sigma^2 = \frac{C^2}{4n}$. By standard sub-Gaussian maximal inequalities, we have

$$\Pr\left(\|g^*\|_\infty \geq C\sqrt{\frac{\log d}{2n}} + \frac{C\delta}{2}\right) \leq 2e^{-\frac{n\delta^2}{2}}$$

for any $\delta > 0$. Setting $\delta = \sqrt{\frac{2\log d}{n}}$ completes the proof of the bound in (12).

Applying the union bound allows us to claim that the guarantees provided by Theorem 1 hold with probability at least $1 - c_0 e^{-n} - \frac{2}{d}$. Guarantees on support recovery follow then by direct application of Corollary 1.

### A.4 Proof to Corollary 3

By (Jain et al., 2014, Lemma 2), we can verify the result of Theorem 1 translates to the matrix case, where the vector $\ell_2$-norm is replaced by the Frobenius norm, the $\ell_1$-norm is replaced by the nuclear norm, and the $\mathrm{HT}_s$ operator is substituted by $\mathrm{PM}_s$.

By applying Theorem 1 and Lemma 2, we conclude that, with probability at least $1 - e^{-c_0 n}$, the iterates satisfy a contractive relation for $\|\Theta_t - \Theta^*\|_F^2$ until the point where $\|\Theta_t - \Theta^*\|_F^2 < \frac{36\|\mathrm{PM}_s(g^*)\|_F^2}{\bar{\mu}^2}$.

To complete the proof, it remains to establish that, with probability at least $1 - \frac{2}{d}$,

$$\frac{36\|\mathrm{PM}_s(g^*)\|_F^2}{\bar{\mu}^2} \leq \frac{7200\sigma^2\zeta(\Sigma)sd}{n\bar{\mu}^2}. \tag{13}$$

Note that $g^* = \frac{1}{n}\varepsilon_i X_i$, by (Wainwright, 2019, Corollary 10.10), we have

$$\Pr\left(\|g^*\|_2 \geq \frac{\lambda_n}{2}\right) \leq 2e^{-2n\delta^2},$$

where $\lambda_n = 10\sigma\sqrt{\zeta(\Sigma)}\left(\sqrt{\frac{2d}{n}} + \delta\right)$.

By setting $\delta = \sqrt{\frac{2d}{n}}$, we have

$$\|g^*\|_2 \leq 10\sigma\sqrt{\zeta(\Sigma)}\sqrt{\frac{2d}{n}},$$

with probability at least $1 - 2e^{-4d}$. We thus apply the union bound to complete the proof of claim (13).

## B  Adaptive Lower Bound

### B.1  Proof of Theorem 2

Let $a_t := \frac{f(\theta_t) - f(\theta^*)}{5\|\mathrm{HT}_s(g_t)\|}$. Suppose the step size $\gamma_t$ used in Algorithm 2 satisfies $\gamma_t = ba_t$ for some scalar $b \in \left[\frac{1}{2}, 1\right]$. By invoking Lemmas 4 and 6, as long as $\|\theta_t - \theta^*\|^2 \geq \frac{36\|\mathrm{HT}_s(g^*)\|^2}{\bar{\mu}^2}$, we have:

$$\|\theta_{t+1} - \theta^*\|^2 \leq \left(1 + \frac{1}{160\bar{\kappa}}\right)\left(1 - \frac{b}{40\bar{\kappa}}\right)\|\theta_t - \theta^*\|^2 \leq \left(1 - \frac{1}{160\bar{\kappa}}\right)\|\theta_t - \theta^*\|^2.$$

This establishes that the iterates exhibit contractive behavior until they enter a small neighborhood of the optimum.

We now consider two possible cases depending on whether the lower bound surrogate $\widetilde{f}_k$ is valid and how the step size compares to $a_t$ during epoch $k$.

Case **(i):** Suppose $\widetilde{f}_k$ is a valid lower bound for $f(\theta^*)$, and that $\gamma_t \leq a_t$ holds for all iterations $t = 0, \ldots, T(\alpha)$ within epoch $k$. Then, the contractive relation applies repeatedly, and we obtain

$$\|\theta_{T(\alpha)} - \theta^*\|^2 \leq \left(1 - \frac{\bar{\mu}}{160\bar{L}}\right)^{T(\alpha)} \|\theta_0 - \theta^*\|^2 \leq (1 + \alpha) \left(1 + \frac{1}{160\bar{\kappa}}\right) \frac{36\|\mathrm{HT}_s(g^*)\|^2}{\bar{\mu}^2}.$$

By the restricted smoothness and strong convexity assumptions, this implies that the function suboptimality satisfies $f(\theta_{T(\alpha)}) - f^* \leq \varepsilon(\alpha)$, thus completing the proof for this case.

Case **(ii):** Alternatively, suppose that $\widetilde{f}_k$ is a valid lower bound, but $\gamma_t > a_t$ for some $t$ in epoch $k$. This condition implies that

$$f(\theta_t) - \widetilde{f}_k > 2(f(\theta_t) - f(\theta^*)),$$

which in turn yields

$$\widetilde{f}_{k+1} = \frac{f(\bar{\theta}_k) + \widetilde{f}_k}{2} \leq \frac{f(\theta_t) + \widetilde{f}_k}{2} < f(\theta^*).$$

Hence, $\widetilde{f}_{k+1}$ is also a valid lower bound. By induction, we conclude that if Case I never occurs, then all $\widetilde{f}_k$, for $k = 1, \ldots, K$, remain valid lower bounds.

Moreover, under this scenario, the sequence $f^* - \widetilde{f}_k$ decreases geometrically. In particular,

$$f^* - \widetilde{f}_{k+1} \leq f^* - \frac{f^* + \widetilde{f}_k}{2} = \frac{f^* - \widetilde{f}_k}{2}.$$

The geometric decrease of $f^* - \widetilde{f}_k$ in case **(ii)** ensures that if case **(i)** never occurs, then there exists some $k_0$ such that $f(\theta^*) - \widetilde{f}_{k_0} < \varepsilon(\alpha)$. In that case, $\bar{\theta}_{k_0}$ is either an output corresponding to case **(i)** (which completes the proof) or an output under case **(ii)**, i.e., it satisfies

$$f(\bar{\theta}_{k_0}) - f(\theta^*) < f(\theta^*) - \widetilde{f}_{k_0} < \varepsilon(\alpha).$$

## C  Other Statistical Guarantees

In this section we provide guarantees for additional statistical models not provided in the body of the paper. The results in Appendix C.1 hold for sparse linear regression. The results in Appendix C.2 apply to some GLMs and require the analysis of the behavior of Sparse Polyak under different regularity conditions. These are not entirely captured in Theorem 1.

### C.1  Sparse Linear Regression

In this section, we assume the dataset consists of data points $z_i = (x_i, y_i)$ for $i = 1, \ldots, n$, where $x_i \in \mathbb{R}^d$ denote the feature vectors, and $y_i \in \mathbb{R}$ denote the responses. The feature vectors are aggregated into the design matrix

$$\mathbb{R}^{n \times d} \ni X \triangleq \begin{pmatrix} x_1^\top \\ \vdots \\ x_n^\top \end{pmatrix}$$

and the responses are aggregated in $\mathbb{R}^n \ni y \triangleq (y_1, \ldots, y_n)^\top$. Let $\theta^*$ denote the ground truth of the statistical model, with $\|\theta^*\|_0 \leq s^*$, and $f^*$ denote the corresponding objective value. Specifically, we assume that the responses $y_i$ and feature vectors $x_i$ are related by $y_i = x_i^T \theta^* + \varepsilon_i$, where $x_i$ are drawn from a $N(0, \Sigma)$ distribution, $\Sigma$ is non singular, $\varepsilon_i \sim N(0, \sigma^2)$, and $x_i$ and $\varepsilon_i$ are i.i.d and independent of one another. Additionally, the objective function $f(\theta) = \frac{1}{2n}\|X\theta - y\|^2$.

From Agarwal et al. (2012)[Lemma 6], it follows that the RSS and RSC conditions hold with probability at least $1 - e^{-c_0 n}$ with coefficients $L = 2\sigma_{\max}(\Sigma)$, $\mu = \frac{1}{2}\sigma_{\min}(\Sigma)$, and $\tau = c_1 \zeta(\Sigma) \frac{\log d}{n}$, where $\zeta(\Sigma) = \max_{i=1,\ldots,d} \Sigma_{ii}$. Here $c_0$ and $c_1$ are universal constants.

**Corollary 4.** *Consider the sparse linear regression problem described above. Let $\{\theta_t\}_{t \geq 0}$ be the sequence of iterates generated by Algorithm 1 or Algorithm 2 when employed to solve a sparse linear*

*regression problem. Suppose that $\widehat{f} = f^*$ for Algorithm 1. Assume we have sufficient samples for $\bar{\mu} > 0$, with $s \geq (240\bar{\kappa})^2 s^*$ for Algorithm 1, and $s \geq (480\bar{\kappa})^2 s^*$ for Algorithm 2. Further, assume that each column of $X$ is $C$-normalized, i.e., $\|\frac{X_j}{\sqrt{n}}\| \leq C$ for $j = 1, \ldots, d$. Here $X_j$ denotes the $j$-th column of $X$. Then, with probability at least $1 - e^{-c_0 n} - \frac{2}{d}$, for any $\alpha \geq \frac{1}{80}$ after $T(\alpha)$ iterations, $\min_{t \leq T(\alpha)} \|\theta_t - \theta^*\|^2$ is upper bounded by*

$$\varepsilon(\alpha) = (1 + \alpha) \frac{288 C^2 \sigma^2 s \log d}{n \bar{\mu}^2}.$$

*The required number of iterations $T(\alpha)$ fulfills:*

$$T(\alpha) \leq \left\lceil \frac{1}{\log\left(1/(1 - 1/80\bar{\kappa})\right)} \log\left( \frac{\|\theta_0 - \theta^*\|^2}{\varepsilon(\alpha)} \right) \right\rceil, \text{ and}$$

$$T(\alpha) \leq \left( 1 + \log_2 \frac{4(f(\theta_0) - f(\theta^*))}{\bar{\mu}\varepsilon(\alpha)} \right) \left\lceil \frac{1}{\log\left(1/(1 - 1/160\bar{\kappa})\right)} \log\left( \frac{\|\theta_0 - \theta^*\|^2}{\varepsilon(\alpha)} \right) \right\rceil,$$

*for Algorithm 1 and Algorithm 2 respectively.*

*Moreover, if $\theta^*$ satisfies the SNR condition* (3)*, we can ensure that after $T(0)$ iterations, the error is upper bounded by $\varepsilon(0)$, and the support of $\theta^*$ has been identified, i.e. $\mathcal{S}^* \subset \mathcal{S}^t \; \forall t \geq T(0)$.*

Corollary 4 establishes the convergence properties of Algorithm 1 and Algorithm 2. The error term is of order $\mathcal{O}\left( \frac{\bar{\kappa}^2 s^* \log d}{n \bar{\mu}^2} \right)$, which is of the same order as that in (Jain et al., 2014, Theorem 3), where a fixed step size is considered under the assumption that $\bar{L}$ is known.

*Proof.* For Algorithm 1, the proof follows the same steps as the proof of Corollary 2. The only difference is the upper bound for $\frac{36\|\text{HT}_s(g^*)\|^2}{\bar{\mu}^2}$, which we provide next.

When the columns of $X$ are $C$-normalized, by (Wainwright, 2019, Example 7.14), with probability $1 - \frac{2}{d}$,

$$\|g^*\|_\infty^2 = \|X^T \varepsilon\|_\infty^2 \leq 8 C^2 \sigma^2 \frac{\log d}{n}.$$

Using the union bound together with (Agarwal et al., 2012, Lemma 6), and the assumptions stated in the Corollary, yield the required assumptions for Theorem 1 to hold. Thus, this completes the proof for Algorithm 1.

As we can see from the proof of Theorem 1 and Theorem 2, the accuracy level $\varepsilon(0)$ is determined by the point at which a lower bound on the step size can be established. According to the formulation of $\gamma_t$ in Algorithm 2, it is guaranteed to be at least half the step size used in Algorithm 1. This observation implies that the accuracy level $\varepsilon(0)$ can also be achieved by Algorithm 2 when the conditions of Case **(i)** in the proof of Theorem 2 are satisfied.

To complete the proof for Algorithm 2, we only need to show that $f(\theta_t) - f^* \leq \frac{18\|\text{HT}_s(g^*)\|^2}{\bar{\mu}}$ implies $\|\theta_t - \theta^*\|^2 \leq \varepsilon(0)$.

By RSC,

$$f(\theta_t) - f^* \geq \bar{\mu}\|\theta_t - \theta^*\|^2 - \|\text{HT}_s(g^*)\| \|\theta_t - \theta^*\|.$$

When $f(\theta_t) - f^* \leq \frac{18\|\text{HT}_s(g^*)\|^2}{\bar{\mu}}$, it implies

$$\frac{\|\text{HT}_s(g^*)\|_s}{\bar{\mu}} \left( \frac{3\|\text{HT}_s(g^*)\|}{\bar{\mu}} + \|\theta_t - \theta^*\| \right) \geq \|\theta_t - \theta^*\|^2.$$

A necessary condition for the inequality above is

$$\|\theta_t - \theta^*\|^2 \leq \frac{36\|\text{HT}_s(g^*)\|_s^2}{\bar{\mu}^2} = \varepsilon(0).$$

$\square$

## C.2 Generalized Linear Models

In this section, we consider a dataset consisting of observations $z_i = (x_i, y_i)$ for $i = 1, \ldots, n$, where $x_i \in \mathbb{R}^d$ denote the feature vectors, and $y_i \in \mathbb{R}$ denotes the corresponding responses. The feature vectors are organized into the design matrix

$$X \triangleq \begin{pmatrix} x_1^\top \\ \vdots \\ x_n^\top \end{pmatrix} \in \mathbb{R}^{n \times d},$$

and the responses are collected in the vector $y \triangleq (y_1, \ldots, y_n)^\top \in \mathbb{R}^n$.

For notational convenience, we let $\theta^*$ denote the true underlying parameter of the statistical model and $f^*$ the corresponding objective function value. We assume the relationship between $x_i$ and $y_i$ is characterized by the conditional distribution

$$\Pr(y_i \mid x_i, \theta^*, \sigma) = \exp \left\{ \frac{y_i x_i^T \theta^* - \psi(x_i^T \theta^*)}{c(\sigma)} \right\},$$

where $\sigma > 0$ is a scale parameter, and $\psi$ is the cumulant function. Given this data generation model, we define the objective function

$$f(\theta) = \frac{1}{n} \sum_{i=1}^n \left( \psi(x_i^T \theta) - y_i x_i^T \theta \right).$$

We assume that $\psi$ is infinitely differentiable with $\psi''(t) > 0$ and uniformly bounded for all $t \in \mathbb{R}$. These assumptions are satisfied in a variety of settings, including logistic regression and multinomial regression Loh and Wainwright (2015). We assume the feature vectors $x_i$ are i.i.d. and drawn from a multivariate normal distribution $N(0, \Sigma)$, where $\Sigma$ is non-singular. Under the setting described above, the RSC condition does not hold everywhere. However, in the described setting it can be shown that the following milder RSC condition holds Loh and Wainwright (2015)

$$f(y) - f(x) - \langle \nabla f(x), y - x \rangle \geq \begin{cases} \frac{\mu}{2} \|y - x\|_2^2 - \frac{\tau}{2} \|y - x\|_1^2, & \text{if } \|y - x\|_2 \leq 1, \quad \text{(14a)} \\ \|y - x\|_2 \left( \frac{\mu}{2} - \frac{\tau}{2} \frac{\|y - x\|_1^2}{\|y - x\|_2^2} \right), & \text{if } \|y - x\|_2 > 1. \quad \text{(14b)} \end{cases}$$

We now present a lemma grouping the results that we need to proceed: the RSS condition, the condition (14), and the order of achievable statistical accuracy for GLMs in our setting. This lemma aggregates results from (Loh and Wainwright, 2015, Proof of Corollary 2, Appendix D.1) and (Negahban et al., 2012, Proposition 2). Notably, while (Negahban et al., 2012, Proposition 2) is originally stated centered only around the ground truth $x = \theta^*$ (c.f. (14)) its proof extends to any given $x$. This implies that while our results are currently stated for $x = \theta^*$, to achieve optimal statistical precision, we can instead state equivalent results to those in Theorem 1 where $\widehat{\theta}$ is such that $f(\widehat{\theta}) = \widehat{f}$.

**Lemma 7.** *For the statistical models described above, with probability at least $1 - c_1 d^{-1} - c_2 e^{-n}$ Assumption 2 and* (14) *hold, and*

$$\|\nabla f(\theta^*)\|_\infty \leq c_0 \sqrt{\frac{\log d}{n}},$$

*where $c_0, c_1, c_2 > 0$ are universal constants. The constants $\mu$, and $L$ in* (14) *and Assumption 2, respectively, depend on $\psi$, and $\Sigma$. Further $\tau = c_3 \frac{\log d}{n}$ where $c_3 > 0$ is a universal constant.*

In the following, we keep the definition $\bar{L} = L + 3\tau s$, and $\bar{\mu} = \mu - 3\tau s$, where $\mu$ is that of (14). Observe that Corollary 5 is stated for Algorithm 1 for simplicity but an analogous statement for Algorithm 2 follows.

**Corollary 5.** *Let $\{\theta_t\}_{t\geq 0}$ denote the iterates generated by Algorithm 1 when applied to the generalized linear models described above. Set the step size rule according to*

$$\gamma_t = \frac{\max\{f(\theta_t) - \widehat{f}, 0\}}{5\|\mathrm{HT}_{2s}(g_t)\|^2}.$$

*Define $R := \|\theta^*\|^2$, and $R_0 := 4R + 1$. Assume we set $\widehat{f} = f^*$, $\theta_0 = 0$, and suppose $s \geq (480 R_0 \bar{\kappa}^2)^2 s^*$. Further, assume the sample size is large enough to ensure $\bar{\mu} > 5 c_0 \sqrt{\frac{2s \log d}{n}}$.*

*Then, with probability at least $1 - \frac{c_1}{d} - c_2 e^{-n}$, we guarantee that for all $t \geq T$ where*

$$T \leq \mathcal{O}\left(\bar{\kappa} \log\left(\frac{n}{s \log(d)}\right)\right) + \mathcal{O}\left(\bar{\kappa}^2 \log(R)\right)$$

*there holds*

$$\|\theta_t - \theta^*\|^2 \leq \left(1 + \frac{1}{160 R_0 \bar{\kappa}^2}\right) \frac{36 c_0 s \log d}{n \bar{\mu}^2}.$$

*Moreover, if $\theta^*$ satisfies the SNR condition (3), we can guarantee that for all $t \geq T$, $\|\theta_t - \theta^*\|^2 \leq \frac{36 c_0 s \log d}{n}$ and that the support of $\theta^*$ has been recovered, i.e. $\mathcal{S}^* \subset \mathcal{S}_t$.*

*Proof.* To establish this result we leverage that the condition (14) combined with Sparse Polyak allow us to establish convergence despite the lack of RSC. More specifically, we will establish that Algorithm 1 exhibits, under the conditions described in the Corollary, at most three modes of convergence:

$$\|\theta_{t+1} - \theta^*\|^2 \leq \begin{cases} \left(1 - \frac{1}{160 R_0 \bar{\kappa}^2}\right) \|\theta_t - \theta^*\|^2, & \text{if } \|\theta_t - \theta^*\| \geq 1, \\ \left(1 - \frac{1}{160 \bar{\kappa}}\right) \|\theta_t - \theta^*\|^2, & \text{if } \|\theta_t - \theta^*\| < 1, \text{ and } \|\theta_t - \theta^*\|^2 \geq \frac{36 c_0^2 s \log d}{n \bar{\mu}^2}, \\ \left(1 + \frac{1}{160 R_0 \bar{\kappa}^2}\right) \frac{36 c_0^2 s \log d}{n \bar{\mu}^2}, & \text{otherwise.} \end{cases}$$
$$(15)$$

Observe that under our current assumptions $\frac{36 c_0^2 s \log d}{n \bar{\mu}^2} < \frac{18}{25}$ and therefore, the list above is exhaustive and the conditions on the second case are compatible.

We begin assuming that $R \geq 1$. We will establish that if the modes of convergence provided above hold for $R \geq 1$, when $R < 1$ we only observe the two last cases.

**(i)** We start off by exploiting (14). Assuming that $\|\theta_t - \theta^*\| \geq 1$ we may follow the strategy in Lemma 4 and exploit that $f$ is convex (but not RSC) to obtain

$$\|\theta_{t+1} - \theta^*\|^2 \leq \left(1 + \sqrt{\frac{s^*}{s}}\right)^2 \left(\|\theta_t - \theta^*\|^2 - 2\gamma_t(f(\theta_t) - f(\theta^*)) + 5\gamma_t^2 \|\mathrm{HT}_{2s}(g_t)\|^2\right). \quad (16)$$

With the choice of step size

$$\gamma_t = \frac{\max\{f(\theta_t) - f(\theta^*), 0\}}{5\|\mathrm{HT}_{2s}(g_t)\|^2},$$

(16) simplifies to:

$$\|\theta_{t+1} - \theta^*\|^2 \leq \left(1 + \sqrt{\frac{s^*}{s}}\right)^2 \left(\|\theta_t - \theta^*\|^2 - \frac{(f(\theta_t) - f(\theta^*))^2}{5\|\mathrm{HT}_{2s}(g_t)\|^2}\right). \quad (17)$$

From Lemma 7 and under the assumption that $\bar{\mu} > 5 c_0 \sqrt{\frac{2s \log d}{n}}$, it follows that with probability at least $1 - c_1 d^{-1} - c_2 e^{-n}$ there holds

$$f(\theta_t) - f(\theta^*) \geq \frac{\bar{\mu}}{2}\|\theta_t - \theta^*\| - \|\mathrm{HT}_{2s}(g^*)\|\|\theta_t - \theta^*\| \geq \frac{\bar{\mu}}{4}\|\theta_t - \theta^*\|,$$

thus by applying the above bound onto (17) we have

$$\|\theta_{t+1} - \theta^*\|^2 \leq \left(1 + \sqrt{\frac{s^*}{s}}\right)^2 \left(\|\theta_t - \theta^*\|^2 - \frac{\bar{\mu}^2}{80\|\mathrm{HT}_{2s}(g_t)\|^2}\|\theta_t - \theta^*\|^2\right). \tag{18}$$

Further, we may upper bound the norm of the gradient as

$$\|\mathrm{HT}_{2s}(g_t)\|^2 \leq 2\left(\|\mathrm{HT}_{2s}(g^*)\|^2_{2s} + \|\mathrm{HT}_{2s}(g_t - g^*)\|^2\right) \tag{19}$$

$$\overset{(i)}{\leq} 2\left(\|\mathrm{HT}_{2s}(g^*)\|^2_{2s} + 2\bar{L}^2\|\theta_t - \theta^*\|^2\right)$$

where in $(i)$ we invoke Lemma 5. Combining (18) with (19) yields

$$\|\theta_{t+1} - \theta^*\|^2 \leq \left(1 + \sqrt{\frac{s^*}{s}}\right)^2 \left(1 - \frac{\bar{\mu}^2}{160\left(\|\mathrm{HT}_{2s}(g^*)\|^2 + 2\bar{L}^2\|\theta_t - \theta^*\|^2\right)}\right)\|\theta_t - \theta^*\|^2. \tag{20}$$

To guarantee that the above implies our first regime of convergence we need to establish that $\|\theta_t - \theta^*\|^2 \leq R$ for all $t \geq 0$. We return to this point after exploring the second and third regime which will be useful in establishing the first.

**(ii)** On the other hand, assume instead that $\|\theta_t - \theta^*\|^2 \leq 1$ and $\|\theta_t - \theta^*\|^2 \geq \frac{36c_0^2 s \log d}{n\bar{\mu}^2}$ then, from (14) the RSC holds and therefore we can invoke the result of Theorem 1 in which

$$\|\theta_{t+1} - \theta^*\|^2 \leq \left(1 - \frac{1}{160\bar{\kappa}}\right)\|\theta_t - \theta^*\|^2. \tag{21}$$

**(iii)** If, instead $\|\theta_t - \theta^*\|^2 < \frac{36c_0^2 s \log d}{n\bar{\mu}^2}$ we have from Theorem 1 that

$$\|\theta_{t+1} - \theta^*\|^2 \leq \left(1 + \frac{1}{160\bar{\kappa}}\right)\frac{36c_0^2 s \log d}{n\bar{\mu}^2} < 1. \tag{22}$$

Observe that from the three cases we consider ((20)- (22)), (21) and (22) already correspond to one of our stated modes of convergence, and thus we are to establish the first.

Clearly, when $R < 1$, $\theta_0$ is in a region in which the RSC holds, and therefore, we will only observe the behavior in (21) and (22). Thus, we are only to prove the first regime of convergence for $R \geq 1$. To establish that the behavior in the first regime holds, we need to establish that $\|\theta_t - \theta^*\|^2 \leq R$ holds for all $t \geq 0$. We proceed to establish this and consequently the behavior in the first regime by induction. Note that by our initial condition $\|\theta_0 - \theta^*\|^2 = \|\theta^*\|^2 = R$ and thus the condition holds for $t = 0$. Suppose $\|\theta_t - \theta^*\|^2 \leq R$ for some $t$. If $\|\theta_t - \theta^*\|^2 < 1$ then either (21) and (22) hold and the proof by induction is complete. If instead, $\|\theta_t - \theta^*\|^2 \geq 1$, (20) holds. Then, by induction hypothesis and under the Corollary's assumption on the sample size there holds

$$\|\mathrm{HT}_{2s}(g^*)\|^2 + 2\bar{L}^2\|\theta_t - \theta^*\|^2 \leq \left(\frac{1}{16} + 2R\right)\bar{L}^2 \leq \frac{R_0\bar{L}^2}{2},$$

thus, combining with (20) and using that by assumption $s \geq (480R_0\bar{\kappa}^2)^2$ we obtain

$$\|\theta_{t+1} - \theta^*\|^2 \leq \left(1 - \frac{1}{160R_0\bar{\kappa}^2}\right)\|\theta_t - \theta^*\|^2 < R.$$

We have thus established the veracity of (15). Observe that (15) together with $\theta_0 = 0$ and $\|\theta^*\|^2 = R$ imply that there exists $t_0 \geq 0$ fulfilling

$$t_0 \leq \lceil -\log(R)/\log(1 - 1/160R_0\bar{\kappa}^2)\rceil$$

such that for all $t \geq t_0$

$$\|\theta_t - \theta^*\|^2 < 1.$$

Further, this implies that in at most

$$\left\lceil \log\left(\frac{36c_0^2 s(1 + 1/(160\bar{\kappa}))\log(d)}{n\bar{\mu}^2}\right)/\log(1 - 1/160\bar{\kappa})\right\rceil$$

additional iterations optimal statistical precision is reached. Finally, if the SNR condition holds, the term $(1 + \frac{1}{160\bar{\kappa}})$ in the third regime is replaced by 1 as a consequence of Corollary 1. $\qquad\square$

Corollary 5 recovers the result in (Loh and Wainwright, 2015, Theorem 3) with the following similarities and differences. To achieve optimal statistical precision, as $\alpha = s\frac{\log d}{n} \to 0$ both Sparse Polyak and (Loh and Wainwright, 2015, Theorem 3) require $\mathcal{O}(\bar{\kappa}\log(\alpha^{-1}))$ iterations. However, we require additional iterations $\mathcal{O}(\bar{\kappa}^2\log(R))$ when $R \geq 1$. We observe however, that our result holds under more general conditions, as we do not make the assumption that $\|\theta^*\| \leq 1$ which is necessary in (Loh and Wainwright, 2015, Theorem 3), where this condition can be relaxed at the expense of requiring the RSC to hold within a larger radius.

We conclude this Appendix by highlighting the fact that for both sparse linear regression and sparse GLMs be verify the rate invariance of Sparse Polyak theoretically. When the problem size increases much faster than the sample size $\frac{d}{n} \to \infty$ but $\frac{s^*\log d}{n}$ and $\Sigma$ remain constant, IHT with Sparse Polyak will reach a $\varepsilon$ neighborhood of the optimal statistical precision within at most $\mathcal{O}(\bar{\kappa}^{-1}\log(1/\varepsilon))$ for linear regression and at most $\mathcal{O}(\bar{\kappa}^{-1}\log(1/\varepsilon)) + \mathcal{O}(\bar{\kappa}^{-1}\log(R))$ when $R > 1$ in the case of GLMs. In both cases, this number does not change with increasing $d$ and $n$. Observe that these results allow us to answer in the affirmative questions **(ii)** and **(iii)** posed in Section 1.

# D   Experiments on real data

**Linear Regression**    We consider a linear regression task using the Large-scale Wave Energy Farm dataset from the UCI Machine Learning Repository Neshat et al. (2020), which is publicly available under the CC BY 4.0 license. The terms of use are described at `https://archive.ics.uci.edu/#terms`. The goal is to predict the total power output of the wave farm based on a sparse linear model. We randomly select 120 samples from the dataset, each containing 149 features. In our experiment, we set the sparsity level to $s = 20$. For the IHT method with a fixed step size, we choose the step size as $8 \times 10^{-12}$. This value is determined via a grid search over the range $[10^{-13}, 9 \times 10^{-12}]$, as step sizes outside this interval result in poor convergence or divergence. The results are presented in Figure 3 **(left)**.

**Logistic Regression**    We evaluate sparse logistic regression using the Molecule Musk dataset Chapman and Jain (1994) from the UCI Machine Learning Repository, which is publicly available under the CC BY 4.0 license. The terms of use are described at `https://archive.ics.uci.edu/#terms`. The task is to classify molecules as musks or non-musks. We randomly select 120 samples from the dataset, each with 166 features. In our experiment, we set the sparsity level to $s = 20$. For the IHT method with a fixed step size, we select a step size of $1.9 \times 10^{-5}$, chosen via a grid search over the interval $[3 \times 10^{-6}, 4 \times 10^{-5}]$. Step sizes outside this range lead to poor convergence or divergence. The results are shown in Figure 3 **(right)**.

We observe that in Fig 3 **(right)** Sparse Polyak performs better than both classic Polyak and IHT with the fixed step-size, even if the step-size is optimized by grid search. This is expected, an adaptive step-size can adapt to the curvature at any point in the algorithm's trajectory, whereas a fixed step-size cannot. We observe that Sparse Polyak performs better than Classic Polyak. On the other hand, in Fig 3 **(left)** we observe that the best fixed step-size performs better than an adaptive step-size. We conjecture that this is due to the factor $1/5$ in Sparse Polyak which we presume is an artifact of our analysis and is not in fact necessary. Despite this fact, sparse Polyak consistently outperforms the classical Polyak rule in the high-dimensional setting.

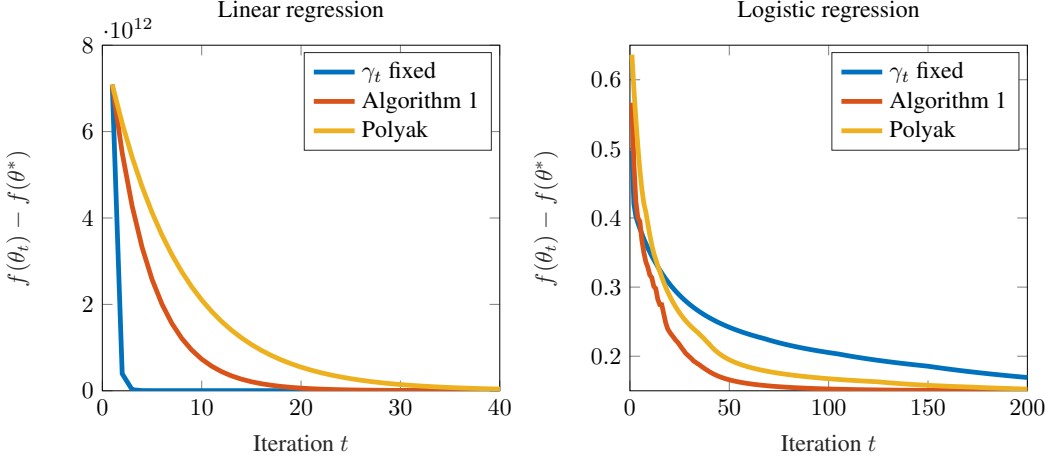

Figure 3: Performance comparison of IHT with optimal constant step size, Sparse Polyak and classical Polyak when performing: **(left)** linear regression on the Wave Energy Farm data set, and **(right)** logistic regression on the Molecule Musk data set.

