# OpenReview forum: "Sparse Polyak: an adaptive step size rule for high-dimensional M-estimation"
_NeurIPS.cc/2025/Conference — NeurIPS 2025 poster_

### Official Review · Reviewer_yL2Y · 2025-06-26

**Clarity:** 4
**Significance:** 4
**Originality:** 3
**Rating:** 5
**Confidence:** 3

**Summary:**

This paper proposes Sparse Polyak, a novel adaptive step-size rule for the Iterative Hard Thresholding algorithm, aimed at solving sparse high-dimensional M-estimation problems, particularly where the dimension $d$ significantly exceeds the sample size $n$. Traditional Polyak step sizes degrade in such settings due to poor estimation of the global Lipschitz constant. The proposed method instead estimates a restricted Lipschitz smoothness constant by focusing on significant descent directions relevant to the sparsity constraint. The authors present a thorough theoretical analysis that establishes the linear convergence of the proposed algorithm under restricted strong convexity and smoothness assumptions, a result comparable to that of IHT with an optimal fixed step size. Notably, the convergence rate only depends on the restricted condition number, which does not grow with dimension in many specific statistical problems, enabling rate invariance under high-dimensional settings. Exact recovery of the support set is established under additional SNR conditions. Experiments are conducted on both synthetic and real datasets to support the theoretical findings.

**Questions:**

1.  Can the authors discuss whether partial or approximate support recovery guarantees could be obtained under relaxed assumptions (e.g., $\Vert \theta^\star_{\min} \Vert$ below the SNR threshold)?
2. Can the authors comment on whether Sparse Polyak could be effective in dense or moderately sparse settings? Will it degrade significantly?
3. How to choose an appropriate $\hat{f}$ in practice? When the loss function is always nonnegative, can we always set it to zero and control the iteration complexity only by stopping criteria?

**Ethical Concerns:**

["NO or VERY MINOR ethics concerns only"]

**Final Justification:**

I have carefully reviewed the authors’ replies, and I find that they have adequately addressed all of my comments and questions. I do not have any further concerns. I would like to retain my original score.

**Limitations:**

Yes

**Quality:**

3

**Strengths And Weaknesses:**

Strengths:
1. The paper provides nontrivial extensions to the convergence theory of IHT with Polyak step-size in the high-dimensional regime. The main theorem and its corollaries are rigorously proved.
2. The experiments are carefully designed and demonstrate the algorithm's advantage over the original Polyak step size, consistency with the optimal fixed step size, and rate invariance under high dimensions, thereby providing sufficient support for the theory.
3. The paper connects its results to classical models, such as sparse logistic regression and low-rank matrix regression, demonstrating broad applicability.

Weaknesses:
1. Corollary 1 requires a relatively strong minimum signal condition for exact support recovery. In many practical applications, signal amplitudes are not well-separated.
2. The paper assumes that each per-sample loss function is convex, which ensures that the empirical risk is convex. However, this assumption excludes a large class of modern M-estimation problems involving non-convex losses (e.g., Tukey’s biweight, logistic loss with non-convex regularizers, or robust statistics). In such settings, the optimization problem becomes non-convex both due to the objective and the constraint, and it is unclear whether the proposed Sparse Polyak step-size remains effective or stable. This limitation restricts the scope of the method to convex loss settings, and the authors do not discuss potential extensions or failure modes under non-convex losses, which are increasingly relevant in robust learning and signal processing applications.

---

> ### Author Rebuttal · Authors · 2025-07-29
>
> We thank the reviewer for their insightful comments and overall positive evaluation of our work. We address the reviewer's concerns in order but merge Weakness 1 and Question 1 for clarity.
>
> $\textbf{Weakness 1 and Question 1:}$ We thank the reviewer for their feedback and question. The minimum signal condition in Corollary 1 is a standard assumption in the literature and has been used prior to claim support identification for IHT-like algorithms. We agree that in some practical scenarios where signal amplitudes are not well-separated, exact support recovery may be more challenging. Upon revision of our proof, we observe that partial support recovery guarantees naturally follow from our analysis. Specifically, any coordinate whose magnitude exceeds the threshold on the right hand side of equation (3) can be correctly recovered, even if the minimum signal condition for exact support recovery is not satisfied. We will clarify this observation in the revision and believe it offers a useful perspective on how our method performs in practice when ideal conditions are not met.
>
> $\textbf{Weakness 2:}$ The reviewer correctly notes that our current analysis assumes convexity of the empirical risk. We agree that extending the proposed Sparse Polyak stepsize to non-convex loss settings--such as robust M-estimation or non-convex regularization (other than $\ell_0$)--is an important and challenging direction. A natural extension would be to consider non-convex problems that satisfy the PL condition. Results for the (standard) Polayk step-size for the PL exist, and are sufficiently fast to guarantee that the expansiveness of the HT operator can be absorbed. For non-convex problems with less structure, sublinear rates are the best one will potentially be able to expect, complicating the absorption of the effect of the HT operator. This will require a more careful analysis of the amount of coordinates that enter/exit the support per iteration. These are two promising directions that we leave as future work.
>
> $\textbf{Question 2:}$ The performance of Sparse Polyak will improve upon that of classical Polyak whenever the restricted condition number is smaller than the condition number (observe that smaller or equal is always true). When restricted strong convexity does not hold, we expect sublinear convergence from both schemes with Sparse Polyak benefiting as the Restricted Lipschitz smoothness constant will be smaller than the Lipschitz smoothness constant. In the case where no sparsity is to be included in the problem (namely gradient descent) both step-size rules collapse and thus yielding the same guarantees but with a slight degradation due to the constant factor in our step size rule.
>
> $\textbf{Question 3:}$ Thank you for the question. While it may seem appealing to use 0 to substitute the value $\widehat{f},$ this approach generally does not guarantee optimal accuracy. The reason is that doing so leads to a more aggressive stepsize which may disrupt convergence altogether. To alleviate the downside of requiring knowledge of $\widehat{f},$ in Appendix B, we desribe a double-loop scheme that adaptively updates a lower bound for $f(\theta^{\star}),$ which substitutes the unknown $\widehat{f}.$ This approach guarantees the same level of accuracy, with an additional computation overhead of at most $\mathcal{O}\left(  \log((\widehat{f} - \widetilde{f}_0)/\varepsilon)\right)$ outer iterations, where $\widetilde{f}_0$ denotes the lower bound, typically set to zero.
>
> Please let us know if further clarifications are needed.

---

> > ### Comment · Reviewer_yL2Y · 2025-08-08
> >
> > Thank you for the detailed response. I have carefully reviewed the authors’ replies, and I find that they have adequately addressed all of my comments and questions. I do not have any further concerns. I would like to retain my original score.

---

### Official Review · Reviewer_drVv · 2025-07-02

**Clarity:** 3
**Significance:** 3
**Originality:** 3
**Rating:** 4
**Confidence:** 2

**Summary:**

The paper considers a high-dimensional statistical estimation problem $\min_{\theta} f(\theta) = \frac{1}{n} \sum_{i=1}^n \ell(z_i,\theta)$ where sparsity of $ \theta $ is at most $s$. They focus on the case when the dimensionality grows much faster than the sample size, i.e. $d/n \rightarrow \infty$. The problem is non convex even if $\ell$ is strongly convex/smooth due to the sparsity constraint. IHT is an algorithm that has been known to efficiently find solutions to this. The algorithm repeatedly applies projected gradient descent where the projection is via a hard thresholding operator. The stepsize used is $1/\overline{L}$ where $\overline{L}$ is the restricted Lipschitz constant, i.e. smoothness constant corresponding to certain important directions.

The paper looks at 3 questions: 1. Do already existing approaches to tune the step size work? 2. Can they achieve the same of better guarantess by choosing an optimal step size? 3. Can they guarantee rate invariance as $d/n \rightarrow \infty$?

The main contribution is an adaptive step size rule that performs well in high-dimensions and maintaines rate invariance. They show simulations that prove existing approaches do not work well in high-dimensions.

Key idea & algorithm
1. Measure smoothness only along sparse directions -- The authors replace the full gradient norm in Polyak’s rule with the norm of the gradient hard-thresholded to the allowed sparsity level s, yielding an estimate of the restricted Lipschitz constant.
2. Sparse Polyak step size  followed by one projected-gradient (Iterative Hard Thresholding) step .

Theoretical guarantees
1. Linear convergence to a neighbourhood whose radius matches the optimal statistical precision, at a rate $O(1/\kappa)$ where $\kappa $is the restricted condition number (Theorem 1) .
2. Rate invariance: if $\kappa$ and the achievable precision stay constant while $ d/n \rightarrow \infty$, the required number of iterations remains constant—matching fixed-step IHT with oracle knowledge of the restricted smoothness constant .

**Questions:**

1. Your theory assumes access to a lower bound on $f(\theta^*)$. In practice you estimate it once per outer loop. How sensitive are convergence and statistical accuracy to misspecifying that bound?
2. Convergence is linear with rate $1/\kappa$. the restricted condition number. Have you observed cases where $\overline{\kappa}$ is large even though sparsity holds, and does Sparse Polyak still help relative to classical Polyak there?

**Ethical Concerns:**

["NO or VERY MINOR ethics concerns only"]

**Final Justification:**

Thanks for the rebuttal and for answering my questions. Based on other reviews, responses, and prior NeurIPS theory papers, I would like to retain my score.

**Limitations:**

Yes

**Quality:**

3

**Strengths And Weaknesses:**

Strengths:

 1. First adaptive step-size rule that retains IHT’s constant-iteration property in the high-dimensional regime.
2. General proof technique that may extend to other adaptive rules.

---

> ### Author Rebuttal · Authors · 2025-07-29
>
> We thank the reviewer for their insightful comments and their overall positive assessment of our work. In what follows we do our best to address the reviewer's questions.
>
> $\textbf{Question 1:}$ Thank you for the question. As noted, our theory assumes access to a lower bound on $f(\theta^{\star}).$ In practice, this may not be known, but 0 is typically a lower bound for $f(\theta^{\star})$ for many learning problems. In Appendix B, we describe a double-loop scheme that relies on such a lower bound, and provide theoretical guarantees (Theorem 2) on both computational cost and accuracy, together with its corresponding proof. Corollaries for specific statistical models can be derived based on Theorem 2 in the same way as the Corollaries in Section 4 are derived of Theorem 1.
>
> Importantly, the final accuracy achieved is not sensitive to the tightness of the initial lower bound. Regardless of how loose the estimate is, the same level of accuracy can still be attained, at the cost of an additional $\mathcal{O}\left( \log ((f(\hat{\theta}) - \tilde{f}_0)/\varepsilon) \right).$ Here $\hat{\theta}$ is a solution to the $s^{\star}$ constrained minimization problem and $\tilde{f}_0$ denotes the lower bound (in general, this can be set to zero).
>
> $\textbf{Question 2:}$ Yes, the reviewer is correct in that convergence is linear and the rate scales with $\bar{\kappa}.$ We agree that when $\bar{\kappa}$ is larger the convergence rate will be slower. In such cases, Sparse Polyak still offers advantages over classical Polyak, particularly due to its sparsity-aware scaling. Specifically, within our set-up the objective function is not strongly convex, and the Lipschitz smoothness parameter (non-restricted) scales with $\mathcal{O}(d/N)$ implying, according to existing theory, potentially sub-linear rates, that will degrade as $d/N$ increases. In short, Sparse Polyak will offer an advantage whenever $\bar{\kappa} < \bar{\kappa}'$ (observe that this always holds in our set-up)  where $\bar{\kappa}'$ denotes the non-restricted condition number, will offer an advantage when the objective function is not strongly convex in all directions, and will yield near (due to the constant factor in our step-size) identical performance when $\bar{\kappa} = \kappa'.$
>
> Please let us know if further clarifications are needed.

---

> > ### Comment · Reviewer_drVv · 2025-08-04
> >
> > I would like to retain my score. Thanks for answering my questions.

---

> > > ### Author Response · Authors · 2025-08-04
> > >
> > > We sincerely appreciate your careful consideration of our responses. Thank you for taking the time to review our work and for your feedback. Please let us know if you have any additional questions.

---

### Official Review · Reviewer_XXZH · 2025-07-02

**Clarity:** 3
**Significance:** 3
**Originality:** 3
**Rating:** 4
**Confidence:** 3

**Summary:**

This paper proposes a sparse adaptation of the classical Polyak stepsize rule for high-dimensional sparse estimation problems. Specifically, the classical Polyak stepsize is given by $\gamma_t=\frac{f(\theta_t)-f(\theta^\ast)}{\|\nabla f(\theta_t)\|^2}$, and the proposed modification replaces the denominator with the squared norm of the top-$s$ hard-thresholded gradient, i.e., $\gamma_t=\frac{f(\theta_t)-f(\theta^\ast)}{\|\mathrm{HT}_s(\nabla f(\theta_t)) \|^2}$. This modification is intended to better align the stepsize with the underlying sparsity structure of the target solution. The paper provides a convergence theorem showing that, under this modified stepsize rule, gradient descent converges linearly to a neighborhood of the optimum. The authors also present numerical experiments to examine empirically the proposed method.

**Questions:**

- Could the authors clarify where the novelty in the convergence proof lies?
- Have the authors considered comparing the proposed method to other sparse optimization methods, such as top-$k$ SGD?
- In practice, how should $s$ be chosen, especially when $s^*$ is unknown? Is there a data-driven or adaptive way to tune this parameter?
- On line 39, “the same” appears to be repeated twice.
- I found the sentence on lines 67–69 a bit unclear. Could the authors consider rephrasing or elaborating on its intended meaning?

**Ethical Concerns:**

["NO or VERY MINOR ethics concerns only"]

**Final Justification:**

The proposal to use an adaptive stepsize, while intuitive, is nonetheless innovative and well-motivated for the sparse optimization setting. The authors provide theoretical convergence guarantees that support their proposed algorithm. Notably, the paper is among the first to emphasize the need for modifying adaptive stepsize rules to address the challenges posed by high-dimensional statistical learning problems, where naive adaptations may degrade as the dimension increases. The authors have also adequately addressed my concerns regarding the lack of comparison to other sparsity-aware methods. Overall, I believe the merits of this paper outweigh the reasons to reject, and I have thus accordingly raised my score.

**Limitations:**

The work is theoretical and methodological in nature and does not appear to raise direct societal concerns.

**Quality:**

3

**Strengths And Weaknesses:**

Strengths:
- The motivation for adapting the Polyak stepsize to sparse settings is clearly explained, and the method is easy to implement.
- Using the top-$s$ hard-thresholded gradient in the stepsize rule is a smart and intuitive adaptation that aligns well with the underlying sparse structure of the problem. It allows the stepsize to focus on informative coordinates and avoids distortion from noise in irrelevant dimensions.
- The authors prove linear convergence for their algorithm. This provides theoretical support for the proposed method in high-dimensional settings.

Weaknesses:
- The paper does not clearly define the original Polyak stepsize rule. For readers who may not be familiar with the literature, it would be helpful to include the classical definition early on to better contextualize the proposed modification.
- While the sparse adaptation is well-motivated, the linear convergence result is somewhat expected from the form of the update. The theoretical result, though important, may not be surprising to readers familiar with sparse optimization theory. While the authors mention that the proof technique may itself be of interest, it is not clear where the novelty lies. It would strengthen the paper if the authors
could more clearly highlight any innovative aspects of the analysis.
- It would be helpful to include comparisons with other sparse-aware baselines, such as top-$k$ SGD, to better contextualize the proposed method's benefits.

---

> ### Author Rebuttal · Authors · 2025-07-29
>
> We thank the reviewer for their insightful comments. We will address the reviewer's concerns in the order they were brought up. However, for both clarity and convenience, we merge the responses to Weakness 2 with Question 1 and Weakness 3 with Question 2.
>
> $\textbf{Weakness 1:}$ Thank you for pointing this out. We agree that including the classical definition of Polyak's stepsize would improve clarity. We will include this in the revision.
>
> $\textbf{Weakness 2 and Question 1:}$ While the sparse adaption we propose is relatively simple in form, and the resulting convergence may appear intuitive, to the best of our knowledge, this is the first adaptive step-size for IHT  that achieves such a result. Not only this, but to our knowledge we are the first to highlight that when dealing with high-dimensional statistical learning problems and employing projected gradient descent, adaptive step-size rules need to be modified to avoid degradation as the ambient dimension $d$ grow. In terms of proof technique, the novelty of our contribution lies in the following:
>
> (i) Our analysis decomposes the dynamics of IHT iterations into the expansiveness of the HT operator and the contractive effect of the gradient descent step. Compared to the original analysis in Jain et. al. 2014 our approach improves clarity and provides a clear understanding of how convergence of IHT is achieved despite not having a fixed step-size.
>
> (ii) We establish that the use of Polyak's step-size is sufficiently large throughout most of the objective function's landscape to guarantee fast convergence, after this stage is completed, we establish that the expansiveness introduced by projection onto the non-convex $\ell_0$ does not harm the overall performance. This aspect is made obvious in the proof of Theorem 1 (Appendix A.1) and discussed briefly in point (iv) in the proof sketch (Section 5). We want to highlight here that Polyak's step-size is typically not used in combination with projected gradient descent over non-convex sets and find that the use of it here yielding optimal statistical guarantees was not necessarily obvious a priori.
>
> $\textbf{Weakness 3 and Question 2:}$ We thank the reviewer for the suggestion. We would like to clarify that a comparison to the classical Polyak stepsize and our sparse variant is already provided in Figure 1, where we demonstrate the invariance of the proposed method.
>
> We agree that comparing to other methods could further contextualize the method's performance. However, our focus in this work is to highlight the need for the design of adaptive step-size rules for the sparse high-dimensional setting. For this, we focused on analyzing a step-size strategy for IHT, which targets an $\ell_0$ constrained optimization problem. In contrast, methods such as top-$k$ SGD address a different class of problems, where the setup, data availability/gradient oracle, and problem geometry differ from those considered in our work. Most importantly, our focus is on problems with explicit sparsity constraints on the solution, which top$-k$ SGD does not enforce. Instead top$-k$ SGD selects the larges $k$ coordinates (in absolute value) with the purpose of approximating the negative gradient direction (this is interesting in the distributed context, as can allow to communicate less). Top-$k$ SGD will therefore be well suited for problems in which the gradient is well approximated by its truncated counterpart, and is thus in a sense, approximately sparse. Observe that in our case, the gradients can be dense (and consequently not well approximated by a sparse quantity) and it's the iterates that are forced to be sparse. We further want to highlight the fact that not controlling for the sparsity of the iterates (as done by top$-k$ SGD) will force the algorithm to traverse the landscape in regions with large Lipschitz smoothness constants potentially implying extremely slow convergence rates. For these reasons, we believe that top$-k$ SGD and related sparsified/quantized gradient methods are not appropriate baselines for direct comparison in our setting. Accordingly, we restricted our empirical evaluations to algorithms using IHT. That said, we acknowledge the importance of comparisons to a broader class of algorithms/formulations and will consider alternative step-size rules (that should be tailed for the high-dimensional regime) in future work.
>
> $\textbf{Question 3:}$ In practice the parameter $s$ is to be chosen by for instance cross validation. We acknowledge that this is not a desirable feature, but it is almost universal in the sparse high-dimensional setting. Either one needs to have a lower bound for $s,$ or knowledge of the noise variance $\sigma^2$ (penalty case). We relax the requirement of IHT of having access to both $s$ and $\bar{L}$ to only $s.$ We want to highlight that this implies that we will be searching for a single parameter as opposed to two as when performing standard IHT.
>
> $\textbf{Question 4:}$ Thank you for pointing this out. We will correct the repetition on line 39 in the revised version.
>
> $\textbf{Question 5:}$ Thank you for mentioning this. The intended meaning of these lines was to convey that the results in Jain et. al. 2014 require the iterate sparsity level to satisfy $s \geq \mathcal{O}(s^{\star}\bar{\kappa}^2).$ This is highlighted here as it plays an important role in highlighting why an adaptive step-size rule is desirable. The scaling $\mathcal{O}(s^{*}\bar{\kappa}^2)$ is only guaranteed in Jain et. al. 2014 if the step size is chosen $\mathcal{O}(\frac{1}{\bar{L}}).$ If the step-size is smaller, the sparsity parameter degrades, implying that the final solution will be unnecessarily dense. Our scheme avoids this issue altogether as the step-size will always be sufficiently large to guarantee that the final solution is sparse in the same sense as in Jain et. al. 2014. We will revise lines 67-80 to clarify this point and enhance readability.
>
> Please let us know if further clarifications are needed.

---

> > ### Comment · Reviewer_XXZH · 2025-08-04
> >
> > I find the authors’ responses to my earlier comments clear and well-explained. I have no further questions and thank the authors for their detailed clarifications.

---

> > > ### Author Response · Authors · 2025-08-04
> > >
> > > Thank you very much for your thoughtful review and for acknowledging our clarifications. Please let us know if you have any additional questions.

---

### Official Review · Reviewer_LwtJ · 2025-07-03

**Clarity:** 3
**Significance:** 3
**Originality:** 3
**Rating:** 5
**Confidence:** 3

**Summary:**

The problem studies minimizing the average of convex losses under the constraint of having at most $s$ non-zero element. The Iterative Hard Thresholding algorithm has been found to a some desirable property called rate invariance: under commonly used statistical models, and as long as the number of samples scales as $s \log d$ (instead of $d$ which is the usual scaling in the dense setting), the number of steps required for achieving optimal statistical precision is independent of $d$ and $n$. This however requires a step size that depends on the restricted Lipschitz smoothness (RSS) constant which is in general unknown to the algorithm. The question then is if existing methods with adaptive step size already achieve this kind of guarantee.

The paper shows empirically that using Polyak’s step size leads to an error that grows with the dimension, thereby ruling out the possibility that the rate invariance property holds in this setting.

Instead, the proposed method  tries to estimate the restricted Lipschitz smoothness. The resulting algorithm is termed sparse Polyak, as it can be viewed as projected gradient descent with a modified version of Polyak’s step size, where the gradient in the denominator is replaced by its hard-thresholded version. The main theorem shows that under the standard assumptions on $f$ used in prior work, convergence is similar to when the RSS constant is known and that the rate invariance property holds. The corollaries specialize the results for specific statistical learning models. The corollaries include logistic regression and matrix regression. The authors also relax some of the conditions in order to get results for generalized linear models for which the original conditions do not hold. The paper also contains numerical experiments that show an advantage of the sparse Polyak method.

**Questions:**

- Does the algorithm work for any possible initialization $\theta_0$ being an $s$-sparse vector?
- If one wants to achieve $f(\theta_t) - \hat{f}$ being upper bounded by a desired value, how do they choose $t$ (this is related to the question about the main theorem in the "Weaknesses" paragraph)? Similar question for the Corollaries.
- For Lemma 1, $\|x_i\| \leq 1$ implicitly means that $\Sigma$ must be upper bounded. Is there a way to relax this condition and possibly have the norm of $\Sigma$ appear in more places in the lemma guarantee?

**Ethical Concerns:**

["NO or VERY MINOR ethics concerns only"]

**Final Justification:**

The authors have provided clear explanations to my points. After reviewing the other rebuttals and discussions my view of the paper remains positive, I thus keep my original score.

**Limitations:**

yes

**Paper Formatting Concerns:**

The main body and appendix were provided in the same pdf

**Quality:**

3

**Strengths And Weaknesses:**

Strengths:
- The paper makes progress on an important problem. Taking advantage of sparsity to avoid performance degrading by the dimension is highly desirable.
-  The main result is general and applicable to many specific problems, as shown in Section 4.
- I liked the evidence shown in Figure 6 of the experiments showing that the adaptive step-size makes more - aggressive progress because it uses local information.

Weaknesses (mostly about writing and presentation):
-  Is question (i) from the beginning of page 2 intended to be a more general question that subsumes (ii) and (iii)? If not, what does (i) ask that is not already covered by (ii) and (iii)?
-  On my first read, I was confused by the description of problem (1) as “statistical,” given that no references to the underlying data distribution were made early on. It would help to clarify from the beginning which quantities (e.g., the RSS parameter) depend on the distribution of the $z_i$ and in what way (e.g., in expectation vs. with high probability). In the current version of the paper the first time these aspects are talked about is in the paper is in Section 4.
- It is stated that the main theorem is almost equivalent to that of Jain et al. 2014. However the theorem there is stated in a different way (it talks about closeness in $f$ values rather than closeness of the parameters). It would be helpful to explain how to translate between the two and to clarify what the additive term $1/(80\kappa)$ corresponds to in the guarantee from Jain et al.
- In Lemma 1, $\hat{f} = f(\theta*)$ is input to the algorithm thus it is implicitly assumed that $f(\theta*)$ is known.


Overall I found some aspects of the presentation confusing but I am overall satisfied with the results of this paper. That said, I am not familiar with the literature, so my opinion is based mostly on the main body of the paper and a quick look at some of the cited works.

---

> ### Author Rebuttal · Authors · 2025-07-28
>
> We thank the reviewer for their insightful comments and their overall positive assessment of our work. We are glad the reviewer liked and it and answer the reviewer's concerns in order:
>
> $\textbf{Weakness 1:}$  We agree with the reviewer in that the questions can be misleading. Questions (ii) and (iii) are subsumed by question (i) in the sense that they concretize in which sense Question (i) is answered in the negative. We will address this in the final version.
>
> $\textbf{Weakness 2:}$ We agree with the reviewer that the distinction between deterministic results and statistical results can be made clearer. To clarify this, we will highlight prior to discussing our results in Section 3 that they are deterministic in nature. We will direct the reader to Section 4 for the statistical setting.
>
> $\textbf{Weakness 3:}$ As the reviewer correctly points out, the deterministic result in Jain et. al. 2014 (Theorem 1) is indeed provided in terms of the objective function value. We highlight that in general, results in terms of estimation error are preferred. To obtain, based on Theorem 1 in Jain et. al. 2014, a result regarding the estimation error (which is preferred in the statistical context) one can exploit the RSC (where here $\bar{\mu}$ includes the terms $\mu,\,\tau,\,s$):
> $$\frac{\bar{\mu}}{2}||\theta_t - \theta^{\star}||^2 + f(\theta^{\star}) + \langle \nabla f(\theta^{\star}), \theta_t - \theta^{\star} \rangle \leq f(\theta_t)\quad (1)$$
> yielding by exploitation of sparsity of $\theta_t$ and $\theta_{\star}$ and Cauchy-Schwarz followed by Young's inequality on the inner product in $(1)$, a result of the form in Theorem 3 in Jain et. al. 2014. Conversely, we can use the RSM to connect the estimation error with the objective function gap
> $$f(\theta_t) - f(\theta^{\star}) \leq \langle \nabla f(\theta^{\star}),\theta_t - \theta^{\star}\rangle + \frac{\bar{L}}{2}||\theta^t - \theta^{\star}||^2$$
> following an analogous approach as for $(1).$
>
> We agree that at face value the addition of the $1/(80\bar{\kappa})$ factor multiplying $\frac{6||HT_s(\nabla f(\hat{\theta}))||^2}{\bar{\mu}^2}$ may appear odd (final precision, Theorem 1), especially given that we can guarantee that the estimation error is contractive as long as its larger than $\frac{6||HT_s(\nabla f (\hat{\theta}))||^2}{\bar{\mu}^2},$ and given that an analogous to this does not appear in Theorem 3 of Jain et. al. 2014. This apparent oddity stems from the adaptive step-size rule. We can guarantee that the adaptive step-size is sufficiently large when we are far away from statistical optimality. This allows to guarantee that the estimation error remains contractive even in the presence of the hard thresholding operator (which can be expansive if the support has not been identified). However, as we get close to statistical precision, we can not guarantee that the step-size remains sufficiently large. This implies that the potential expansive effects of the projection onto the non-convex set can not be controlled by the gradient step, and we may exit the ball of radius $\frac{6||HT_s(\nabla f(\hat{\theta}))||^2}{\bar{\mu}^2}$ by a factor at most $1/(80\bar{\kappa}).$ If this happens, we contract back and thus the guarantee provided holds. This discussion is partially reflected in (iv) in the proof sketch (Section 5) and its formal counterpart can be found in the proof of Theorem 1 in Appendix A.1. Observe also that as a consequence, this term does not appear when support recovery is possible due to sufficiently high SNR. This is because if the support does not change, the HT operator does not introduce expansion.
>
> $\textbf{Weakness 4:}$ We specify the results for the statistical models under the assumption that $f(\theta^{\star})$ is known. We provide the analogous result to Theorem 1 when $f(\theta^{\star})$ is unknown in Theorem 2 (Appendix B). Here, we do not require knowledge of $f(\theta^{\star})$ but a lower bound to it. For statistical learning problems, 0 is generally a valid lower bound, i.e. $0 \leq f(\theta^{\star}).$ In this case, optimal statistical precision is achievable at the price of computing additional $\log(1/\varepsilon)$ factor in terms of number of iterations. The analogous results to those in Section 4 in the case where $f(\theta^{\star})$ is unknown can then be derived as Corollaries to Theorem 2 in the same way as the results in Section 4 are corollaries of Theorem 1.
>
> $\textbf{Question 1:}$ The reviewer is correct in that the algorithm will converge as long as $\theta_0$ is s-sparse. One may select $\theta_{-1}$ to be an arbitrary vector and set $\theta_0$ to include only its $s$ largest elements (in absolute value) and zero elsewhere,  or alternatively, and more conveniently (unless prior knowledge on where the solution may lie is) one may set $\theta_0 = 0.$
>
> $\textbf{Question 2:}$  This is a good point and we acknowledge that setting a practical termination criteria is non trivial. If an upper bound to the quantity $||\nabla f(\hat{\theta})||^2_{\infty}$ is known (in terms of order this is not unreasonable, i.e. for $\hat{\theta} = \theta^{\star}$ $||\nabla f(\hat{\theta})||^2_{\infty} \leq \mathcal{O}(\frac{\log d}{N})$) and therefore from $(1)$ it follows that
>
> $$||\theta^t - \theta^{\star}||^2 \leq \frac{1}{2\bar{\mu}}\left(f(\theta^t) - f(\theta^{\star})\right) + \frac{s}{2\bar{\mu}}||\nabla f(\theta^{\star})||^2_{\infty}.$$
>
> Thus, setting $\varepsilon = c s ||\nabla f(\theta^{\star}) ||^2_{\infty}$ (which is known up to a constant factor $c$) and requesting $f(\theta_t)-f(\hat{\theta}) \leq \varepsilon $ guarantees an upper bound on the estimation error of the order of optimal statistical precision. We note that the total number of iterations this will require is given by Theorem 1. Observe that this explanation holds for all statistical models as they are all based on the same deterministic result.
>
> We want to highlight that one can not establish that the objective function value decreases monotonically as in standard gradient descent when the step-size is chosen to be sufficiently small. This is because Polyak's step size can be larger than that yielding faster convergence in practice but a loss of monotonicity. Further, in the standard (low-dimensional) scenario, with strongly convex functions and Lipschitz smooth gradients, if we were to employ Polyak's step-size, the termination criteria $f(\theta_t) - f^{\star} \leq \varepsilon$ guarantees $\frac{\mu}{2}||\theta_t - \theta^{\star}||^2 \leq \varepsilon,$ where $\mu$ is unknown. This implies that even in the standard case, we have a sense of the precision at the level of the iterates except for a constant factor.
>
>
> $\textbf{Question 3: }$ This an important point. We agree that the assumption $\|x_i\|\leq 1 $ is extremely restrictive. In fact, this will in general not hold in the high dimensional setting.  We included this result as is the one present for logistic regression among works considering the Hard Thresholding algorithm and thus facilitates comparison to prior works. This restriction originates from Corollary 1 of Yuan and Li 2021, where it was used to ensure that the RSC condition holds. This is necessary because GLMs generally do not satisfy the RSC without additional structural constraints. In contrast, our analysis relies on a milder version of the RSC condition (Loh and Wainwright 2015) allowing us to avoid such restrictive assumptions. This and guarantees for our algorithm's convergence are stated and proven in Appendix C.2, as a function of the covariance matrix $\Sigma$. We highlight that this is the first IHT (regardless of step-size) result to work for logistic regression without artificial assumptions on the statistical model. This is because Polyak's step-size can guarantee linear convergence under milder conditions on the lower bound on the function's curvature.
>
> Please let us know if further clarifications are needed.

---

> > ### Comment · Reviewer_LwtJ · 2025-08-05
> >
> > Thank you for the detailed response. I went through the response and it addresses all of my points and questions. I do not have further questions for the authors.

---

### Decision · Program_Chairs · 2025-09-17

**Decision:**

Accept (poster)

**Comment:**

**Summary:** The paper studies a variant of the Polyak step-size for high-dimensional statistical estimation problems, where $d >> n$. The paper shows that using the Polyak step-size leads to an error that grows with the dimension. Instead, the proposed method tries to estimate the restricted Lipschitz smoothness. The resulting algorithm is termed sparse Polyak, as it can be viewed as projected gradient descent with a modified version of Polyak’s step size, where the gradient in the denominator is replaced by its hard-thresholded version. The main theorem shows that under the standard assumptions used in prior work, convergence is similar to when the RSS constant is known. The corollaries specialize the results for specific statistical learning models, and include logistic regression and matrix regression. The paper also contains numerical experiments that show an advantage of the sparse Polyak method.

**Strengths:**
- Well-written paper, with a clear contribution and a novel modification to a classical method.
- Strong theoretical results showing linear convergence in high-dimensional settings.
- Carefully designed experimental evaluation to validate the theory

**Weaknesses:**
- Needing to choose the sparsity level $s$ via cross-validation is not ideal.

**Decision and Suggested Changes:** The reviewers unanimously agree that the paper is well-written, and its contributions merit acceptance. After carefully reading the rebuttal/discussion, I tend to agree. Please incorporate the reviewer comments in the final version of the paper. In particular, addressing the following concerns will help strengthen the current version of the paper:
- Explanation of the comparison to Jain et al, 2024 and discussion about the termination criterion (response to Rev. LwtJ)
- Clarification about the comparison to classical Polyak (response to Rev. drVv, Rev. yL2Y)